# Extending quantum-mechanical benchmark accuracy to biological ligand-pocket interactions

Mirela Puleva [1,2], Leonardo Medrano Sandonas [1,3] ✉, Balázs D. Lőrincz[4,5,6], Jorge Charry [1,7], David M. Rogers[8], Péter R. Nagy [4,5,6] ✉ & Alexandre Tkatchenko [1,2] ✉

Predicting the binding affinity of ligands to protein pockets is key in the drug design pipeline. The flexibility of ligand-pocket motifs arises from a range of attractive and repulsive electronic interactions during binding. Accurately accounting for all interactions requires robust quantum-mechanical (QM) benchmarks, which are scarce for ligand-pocket systems. Additionally, disagreement between "gold standard" Coupled Cluster (CC) and Quantum Monte Carlo (QMC) methods casts doubt on many benchmarks for larger non-covalent systems. We introduce the "QUantum Interacting Dimer" (QUID) benchmark framework containing 170 non-covalent (non-)equilibrium systems modeling chemically and structurally diverse ligand-pocket motifs. Symmetry-adapted perturbation theory shows that QUID broadly covers non-covalent binding motifs and energetic contributions. Robust binding energies are obtained using complementary CC and QMC methods, achieving agreement of 0.5 kcal/mol. The benchmark data analysis reveals that several dispersion-inclusive density functional approximations provide accurate energy predictions, though their atomic van der Waals forces differ in magnitude and orientation. Contrarily, semiempirical methods and empirical force fields require improvements in capturing non-covalent interactions (NCIs) for out-of-equilibrium geometries. The wide span of NCIs, highly accurate interaction energies, and analysis of molecular properties take QUID beyond the "gold standard" for QM benchmarks of ligand-protein systems.

Accurate computational modeling of physicochemical phenomena in protein-ligand systems is vital for accelerating the early stages of the drug development pipeline[1–4]. Reliable and "clean room" experimental measurements of the binding affinity can be costly due to multiple factors, e.g., dissolved electrolytes, solvent concentration, and target protein misfolding[5], which makes robust computational methods crucial for improving efficiency and gaining detailed mechanistic understanding into the ligand-protein systems[6]. Understanding and consequently controlling non-covalent interactions (NCIs) in a targeted way can aid the compound design process to achieve optimal target selection[7]. Hence, an in-depth description of NCIs—dominant interactions determining structural configuration and ligand-pocket binding mechanism—is indispensable for binding affinity simulations. Accurate calculations are indeed critically important as even errors of 1 kcal/mol can lead to erroneous conclusions about relative binding affinities[8].

Molecular mechanics (MM) force fields (FF), along with docking and free-energy methods, have been widely used due to their

---

affordable computational cost for estimating structural and thermo-dynamical properties of complex (bio)molecular systems[9,10]. Despite significant progress in the development of accurate and polarizable MMFFs[11,12], most of them treat ubiquitous non-covalent polarization and dispersion interactions using effective pairwise approximations, often resulting in inaccuracies or lack of transferability between different chemical subspaces, both being prerequisites for accurate novel drug predictions[13,14]. Meanwhile, a broad range of quantum-mechanical (QM) methods have become available with different trade-offs between accuracy and size spanning from the less expensive but more approximate semiempirical (SE)[15,16] approaches, passing through the density functional theory (DFT)[17–25] ones to the "gold standard" Coupled Cluster (CC)[26] and Quantum Monte Carlo (QMC)[27–29] methods. However, achieving a sufficiently reliable and reproducible QM description of NCIs remains computationally prohibitive for realistic ligand-pocket systems, preventing further development of accurate and efficient free-energy simulation methods as well as enhanced mechanistic models for ligand-protein design.

Aiming to close this gap and improve our understanding of ligand-pocket binding, here we develop the "Quantum Interacting Dimer" (QUID) benchmarking framework. At this first stage, QUID contains 170 chemically diverse large molecular dimers (42 equilibrium and 128 non-equilibrium) of up to 64 atoms, including the H, N, C, O, F, P, S, and Cl chemical elements, encompassing most atom types of interest for drug discovery purposes. The selection of ligand-pocket motifs is accomplished through exhaustive exploration of different binding sites of nine large flexible chain-like drug molecules from the Aquamarine dataset[30] systematically probed with a small monomer, once with benzene ($C_6H_6$) and once with imidazole ($C_3H_4N_2$). Eight non-equilibrium conformations are also generated per each for a selection of 16 equilibrium dimers, sampling along the non-covalent bond dissociation direction. Given the structural and chemical diversity of the resulting conformations, a single dimer can exhibit multiple types of steric effects and NCIs simultaneously, including, but not limited to, polarization, $\pi$–$\pi$ stacking, hydrogen and halogen bonds.

The design process of QUID was inspired by several landmark QM datasets. Most of them are focused on DFT-based physicochemical properties of single molecules up to a few hundred atoms[30–34]. Only a limited number investigate NCIs, via the interaction energies ($E_{int}$), in molecular systems at the benchmark ab initio level of CCSD(T)/CBS. Among them, one can find the well-established S22[35,36] and S66(x8)[37,38] datasets; L7 with a few specific larger systems[39], as well as the newer NENCI[40], DES370K[41], and SAPT10K[42] ones with improved chemical diversity. Specifically for modeling ligand-pocket interactions, a recent dataset Splinter[43] has been developed—in it two distinct small molecules represent common fragments in proteins and small-molecule ligands. While Splinter features charged monomers and good chemical diversity, its compounds are all of similar size, up to ≈40 atoms, thus offering limited venue for reproducing size-dependent NCIs or geometric arrangements typical of ligands in a pocket. Furthermore, large QM datasets of energies and atomic forces have been generated for non-covalent systems by combining structural data from MMFF-based simulations with DFT methods[44–47] to develop more robust machine-learned FFs for (bio)molecular simulations.

Our QUID framework aims to redefine the state-of-the-art in benchmarking NCIs in complex molecular systems. First, we define a "platinum standard" for ligand-pocket interaction energies (not to be confused with the platinum standard as used in the quantum chemistry community). It is obtained by establishing a tight agreement between two completely different "gold standard" methods for solving the Schrödinger equation: LNO-CCSD(T) and FN-DMC, thereby largely reducing the uncertainty in highest-level QM calculations. Second, the analysis of interaction components allows us to describe a wide range of NCIs of relevance to ligand-protein systems. Third, a comprehensive analysis of approximate empirical, semiempirical, and first-principles

calculations allows us to pinpoint improvements required in each of these methodologies to move towards trustworthy free-energy simulation methods. We suggest that only a comprehensive combination of such benchmark analyses enables an unbiased understanding of NCIs in realistic molecular complexes as illustrated in this work for model ligand-pocket systems.

## Results

### Quantum-mechanical exploration of binding interactions

Modeling the NCIs of a ligand in a protein pocket is essential for determining the structural arrangements of natural enzyme substrates and drug candidates. This is why the QUID model systems, which represent the most frequent ligand-pocket interaction types, were created to investigate the impact of adequately describing NCIs on binding features and the influence of structural binding conformations on electronic properties. Each QUID dimer comprises a large monomer as a host and a small monomer representing a ligand motif. To achieve a proxy model representation of the interactions on the pocket-ligand surface, the large monomers are chosen from chemically diverse drug-like molecules of ≈50 atoms, with flexible chain-like geometry allowing for folding and multiple accessible binding sites (aromatic rings) (see Fig. 1). In doing so, nine molecules (including C, N, O, H, F, P, S, and Cl atoms) that met our criteria were extracted from the Aquamarine dataset[30]. Two small monomers were selected to represent the ligand interactions: benzene, the quintessential aromatic compound present in the phenylalanine side-chain, and imidazole, present in histidine, a more reactive and also a commonly used drug motif[48]. The resulting complexes represent the three most frequent interaction types appearing on the pocket-ligand surface, that is aliphatic-aromatic, H-bonding, and $\pi$-stacking, which are found in more than 100,000 interactions within PDB structures[49]. The QUID dimers are comprised of monomers interacting in one or more of the aforementioned ways, with many presenting non-covalent effects of mixed character, e.g., combining $\pi$-stacking and H-bonding. In each initial dimer conformation, the aromatic ring of the small monomer was aligned with that of the binding site at a distance of 3.55 ± 0.05 Å (similar to S66 dimers[37]), and the dimer was then optimized at the PBE0+MBD level of theory.

Post-optimization, we obtained 42 QUID equilibrium dimers that were split into three categories based on the structural shape of the corresponding large monomer: 'Linear', in which the original chain-like geometry is mainly retained; 'Semi-Folded', in which parts of the large monomer are bent while other sections remain linear, and 'Folded', in which the big monomer encapsulates the smaller one. Thus, a variety of pockets with different packing densities is modeled by the QUID dimers, e.g., the folded F1I3 mimicking a more crowded binding pocket[50], or the linear L2B1 representing a toy model of a more open surface pocket[51]. This classification is shown in terms of the radius of gyration and chemical diversity in Supplementary Fig. S1 of the Supplementary Information (SI). As a result, a wide range of interaction energies $E_{int}$ between the monomers is produced, ranging from −24.3 to −5.5 kcal/mol at PBE0+MBD level, with imidazole usually resulting in stronger non-covalent bonds (see Supplementary Fig. S2). The prediction of the $E_{int}$ values will be investigated in "Assessing the performance of DFT, semiempirical, and empirical methods" section with diverse computational methods.

Next, a representative selection of 16 dimers is used to construct non-equilibrium conformations along the dissociation pathway of the non-covalent bond (along $\pi$–$\pi$ or H-bond vector), modeling snapshots of a ligand binding to a pocket. These conformations were generated at eight distances, characterized by a multiplicative dimensionless factor $q$, defined as the ratio of the inter-monomer distance to that of the equilibrium dimer. The chosen values of $q$ are 0.90, 0.95, 1.00, 1.05, 1.10, 1.25, 1.50, 1.75, and 2.00, where $q = 1.00$ denotes the equilibrium dimer. The structure of these non-equilibrium dimers was also optimized at PBE0+MBD level with the heavy atoms of the small monomer

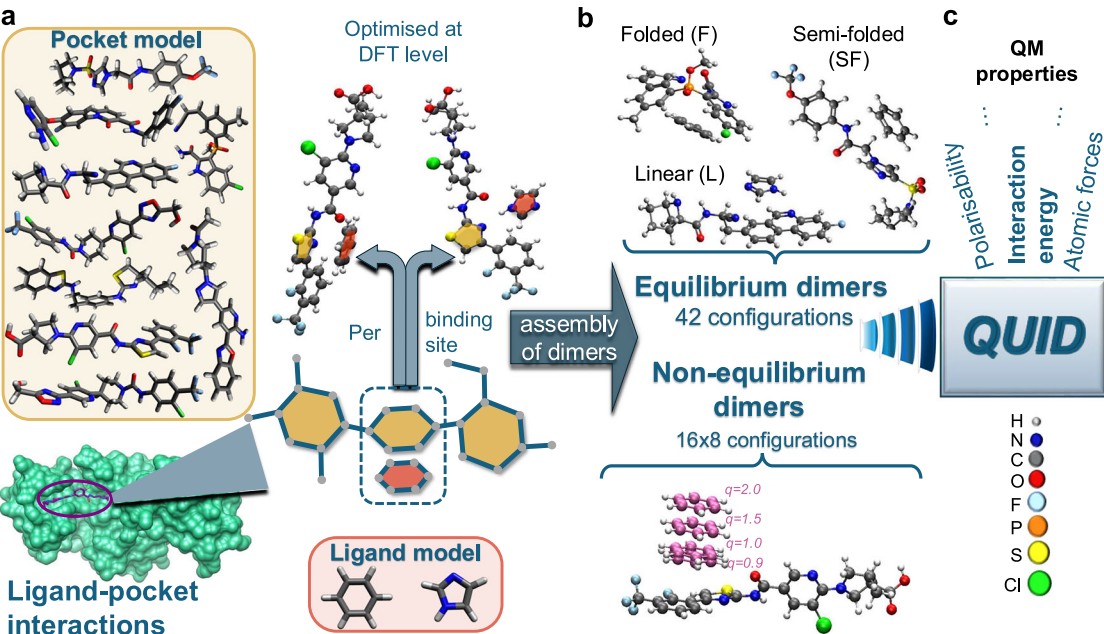

**Fig. 1 | Schematic representation of the generation of the QUantum Interacting Dimer (QUID) dataset.** In (**a**), the molecules forming a QUID structure modeling a protein pocket and those modeling a ligand are presented at the top and right side of a protein-ligand complex[121] visualized with ChimeraX[122]. Each dimer is composed of one of nine big monomers containing multiple potential binding sites and a small monomer binding to one of them. The resultant dimer arrangement is optimized at Density Functional Theory (DFT) level (PBE0+MBD). The resulting conformer geometries are shown in (**b**), categorized as Linear, Semi-Folded, and Folded. For 16 equilibrium dimers, eight non-equilibrium conformations are designed along the dissociation of the non-covalent bond as illustrated by an example. $q$ is a multiplicative dimensionless factor in the range of 0.9 to 2, which denotes the ratio of the inter-monomer distance to that of the equilibrium dimer. In (**c**), a graphic summary of the QUID dataset with its chemical composition and some available QM (quantum mechanical) molecular properties is shown.

and the respective binding site kept frozen. The resulting systems demonstrate the varied $E_{int}$ spectrum for different pocket types via the structure categories in equilibrium and along the dissociation paths (see Supplementary Fig. S2). The generation protocol for the 42 equilibrium and 128 (16 × 8) non-equilibrium dimers is schematically outlined in Fig. 1 and detailed in the "Methods" section. These model systems represent a significant step forward in accurately investigating ligand-pocket interactions, characterized by robustly optimized molecular dimers that exhibit chemical diversity, larger size, and complex binding conformations.

## Analysis of non-covalent interaction components

A detailed characterization of the physical interactions in ligand-pocket systems is needed to both aid understanding and ensuring diversity in the coverage of interactions. To that end, a decomposition of $E_{int}$ of QUID systems with SAPT analysis was performed, specifically with the sSAPT0 version due to its good balance between accuracy and computational cost[52], which is a slightly modified recipe that involves a rescaling of $E_{exch-disp}^{(20)}$ and $E_{exch-ind}^{(20)}$ terms based on an empirically adjusted proportion between $E_{exch}^{(10)}$ and $E_{exch}^{(10)}(S^2)$. The sSAPT0 $E_{int}$ predictions for the equilibrium dimers were found qualitatively consistent with those computed at the LNO-CCSD(T) level (MAE of 0.85 kcal/mol). The largest discrepancies are found for dimers with imidazole as the small monomer and with both π−π stacking and H-bonding contributions to the non-covalent interaction. The most pronounced discrepancy occurs for the folded F1I1 dimer, with a value of 1.97 kcal/mol. Case-by-case results for all equilibrium QUID dimers are shown in Supplementary Fig. S3. The SAPT analysis provides insight into the energy components of the NCIs, namely induction, dispersion, electrostatic, and exchange contributions, which elucidate the balance of intermolecular interactions. This measure of dispersion and electrostatic components has been used before to gain insight into specific protein-ligand interactions[42,53] and will provide a solid basis for

interpreting the different predictions of $E_{int}$ from different QM methods.

The variety in the spread of the sSAPT0 components for the QUID equilibrium dimers is shown in a stacked histogram in Fig. 2a. This showcases the diversity as a result of the different chemical environments and NCI types (see the associated NCI-plots[54] in Supplementary Fig. S4). Notably, the dispersion and electrostatic terms are strongly represented, while the induction contribution is the smallest, about 15−20% of the total value of $E_{int}$. This indicates a supplemental role of the polarization effects on one monomer as a result of permanent dipoles of the other, with the imidazole-based dimers consistently having larger induction components compared to their benzene counterparts when settled in the same environment. For 9 dimers, electrostatics are the dominant term (see Supplementary Fig. S5), being particularly strong for the SF1I2 dimer. Such strong electrostatics are rare in the QUID systems as seen in Fig. 2b, where single outliers are seen for electrostatic contributions higher than −14 kcal/mol, while there is a skew towards smaller values in the range −8.0 to −4.0 kcal/mol. All 9 electrostatic-dominant cases involve imidazole as the small monomer. This is consistent with the presence of the two N atoms and their lone pair orbitals, capable of forming H-bonds and dipole−dipole interactions. For SF1I2, also a sulphonyl group is found near the binding site (see Supplementary Fig. S4), and the amino H atom in imidazole is interacting with the $SO_2$ functional group, possibly forming an H-bond, in addition to the second H-bond between the imidazole and an amino group in the large monomer. Dispersion is the dominant component for the other 33 equilibrium dimers, as expected given the choice of binding sites (pre-optimization) on accessible aromatic rings arranged for π−π stacking. The spreads of the electrostatic and dispersion components are contrasted against the $E_{int}$ spread in Fig. 2. $E_{int}$ ranges from −8 to −20 kcal/mol, clearly larger than π−π stacked or single H-bonds in small

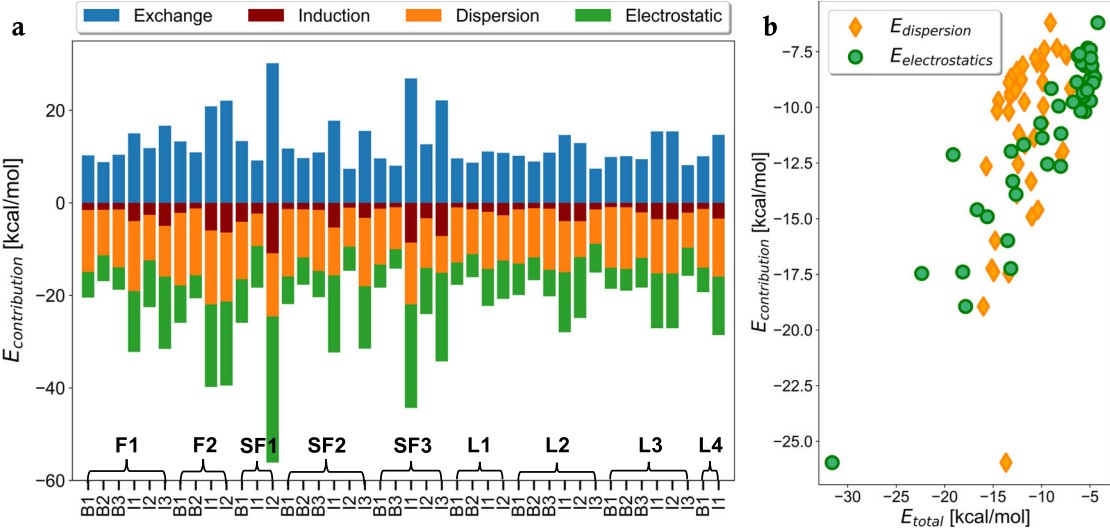

**Fig. 2 | Symmetry-adapted perturbation theory (SAPT) energy decomposition analysis. a** The Exchange, Induction, Dispersion, and Electrostatic contributions to the interaction energies $E_{int}$ of all 42 equilibrium QUantum Interacting Dimer (QUID) systems are depicted for the results obtained with SAPT level sSAPT0/ jaDZ[52]. The dimer names on the x axis can be read as the bracket first half followed by the tick label, e.g., F1 followed by B1 for the F1B1 dimer first on the x axis. **b** Scatter plot of the Dispersion and Electrostatics contributions compared to the total interaction energies is shown for all 42 equilibrium QUID dimers.

dimers, which usually contribute between −2 to −8 kcal/mol to the interaction energy[37]. Furthermore, the dispersion component around −10 kcal/mol is larger than that obtained in a benzene dimer around −5 kcal/mol[55]. Some dimers can also depict mixed NCI character, e.g., both an H-bond and $\pi$−$\pi$ stacking in F1I3, or both sandwich $\pi$−$\pi$ stacking and T-shaped interaction of aromatic groups simultaneously in F1I1. Therefore, we can conclude that the ligand binding is enhanced due to the collective interactions with the pocket with SAPT characterizing the specifics of those interactions and also revealing the complexity of the physical interactions of the QUID systems.

### Benchmarking the quantum-mechanical benchmarks towards "platinum standard"

Reliable models for pocket-ligand systems rest on robust methods performing consistently and accurately in such systems. A prerequisite towards understanding and estimating the performance of current methods is the existence of reliable data, which can be challenging when results in literature obtained at "gold standard" level of computation have been found to disagree[56,57], and comprehensive and computationally expensive studies are needed to explore sources of the discrepancy[58]. Hence, to establish a thoroughly dependable reference for the interaction of ligands with a protein pocket, $E_{int}$ for the QUID proxy systems has been obtained and compared for the two gold standard methods LNO-CCSD(T)[59–62] and Fixed-Node Diffusion Monte Carlo (FN-DMC)[27–29] to produce a "platinum standard". Within our methodology, both approaches were employed with particular care to achieve convergence, see the "Methods" section for more details. The results were compared for the equilibrium set of QUID dimers and found in agreement within the uncertainty estimates of the two reference quantum methods in 31 of the 42 cases (i.e., 74%) as seen in Fig. 3. The MAE between the two methods is 0.47 kcal/mol compared to 0.38 kcal/mol mean absolute value of the uncertainty estimate for both FN-DMC and LNO-CCSD(T), respectively. The benchmark ab initio methods are in good agreement for the QUID systems, a result previously found unattainable for larger non-covalent systems with dispersion-dominated interactions, e.g., in the L7 dataset[56,63]. In QUID, let us take as an example one case (SF3I3) out of the studied ones, where the LNO-CCSD(T) prediction with its uncertainty estimate lies outside one-sigma agreement with the FN-DMC result but remains

within two sigmas. This still means statistical consistency considering 68% and 95% assigned to one and two sigma intervals, respectively.

Analysis of the discrepancy patterns between FN-DMC and LNO-CCSD(T) was performed by assessing the character of the non-covalent bonding from sSAPT0/jaDZ. The results were found to be consistent with a recent study performed on the S66 dimers dataset[57]—in both cases the results indicate that the dominant electrostatic component of the interaction energy correlates with the disagreements between the gold benchmark methods. Details are given in Fig. 3b, presenting a plot of the difference in the interaction energy predictions versus a log of the ratio of the electrostatic and dispersion sSAPT0 components. For the QUID systems, all the cases of disagreement involve a H-bond between the monomers, although some dimers with H-bonds in the interaction with higher dispersion contribution were found to be in agreement. These results are in line with the 0.9 kcal/mol deviation found by Shi et al.[57] for the acetic acid dimer. From this perspective, the QUID systems differ from the supramolecular complexes with extended $\pi$−$\pi$ interactions, where some considerable disagreements between CCSD(T) and FN-DMC were uncovered[56]. As noted in ref. 56 and recent studies[57,64–66], the FN approximation, time-step discretization, pseudopotential, post-CCSD(T) terms, basis set extrapolation used in CCSD(T), and other high-order effects could be notable for extended $\pi$−$\pi$ interactions. However, beyond CCSD(T) corrections are deemed to be very small for our purposes in H-bonded dimers[64,66]. Hence, based on these studies and our comparison between "gold standard" methods, we take LNO-CCSD(T) as a practical and reliable reference for $E_{int}$ of ligand-pocket NCIs in the complex QUID dimers. LNO-CCSD(T) results were subsequently obtained for all 42 equilibrium dimers and the full dissociation curves of a representative selection of 6 dimers (details in "Methods" section).

### Assessing the performance of DFT, semiempirical, and empirical methods

Given the "platinum standard" $E_{int}$ reference, we next conduct a comprehensive and reliable examination of its prediction and reliability obtained from different approximate computational methods for capturing NCIs in QUID equilibrium systems. This is done with the goal of identifying approximate methods that can be used in eventually building a trustworthy pipeline for calculating binding affinities.

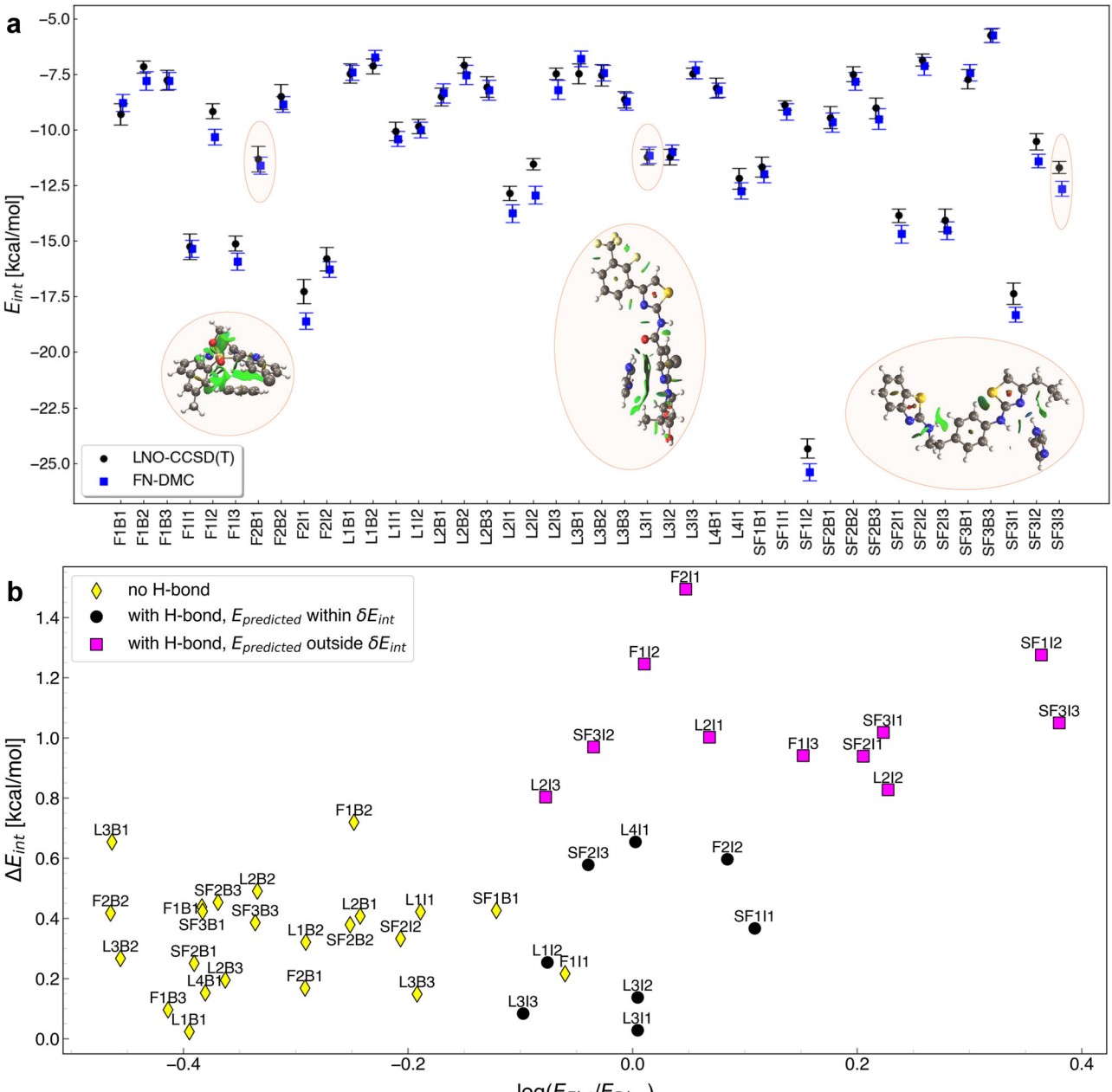

**Fig. 3 | Towards "platinum standard" in $E_{int}$ by benchmarking "gold standard" methods LNO-CCSD(T) and FN-DMC. a** Comparison of the interaction energies ($E_{int}$) computed using Fixed-Node Diffusion Monte Carlo, FN-DMC (0.015 or 0.025 time step) and Local Natural Orbitals - Coupled Cluster with Singles, Doubles, and perturbative triplets, LNO-CCSD(T) (extrapolated to Complete Basis Set (CBS) and Local Approximation-Free (LAF) limit) for the 42 equilibrium QUantum Interacting Dimer (QUID) dataset. For FN-DMC, error bars represent estimated one-$\sigma$ statistical error for $4 \times 10^8$ configurations. For LNO-CCSD(T), the error bars correspond to the estimated uncertainty from the best CBS and LAF extrapolations. Three cases are highlighted: L3I1 for which the methods are in perfect agreement, F2B1 for which the methods agree within their uncertainty estimates, and SF3I3 as the one case for

which they are in slight disagreement. The NCI plots illustrating the non-covalent interactions in those molecular dimers are also shown[54]. **b** Scatter plot of the absolute differences in the prediction of the interaction energies between LNO-CCSD(T) and FN-DMC, $\Delta E_{int}$, versus the log of the ratio between the electrostatic (Elst) and dispersion (Disp) Symmetry Adapted Perturbation Theory (SAPT) components from sSAPT0[52]. The equilibrium QUID dimers are divided into three subsets: yellow (no H-bond in non-covalent interaction) and black (H-bond in non-covalent interaction) symbols indicate cases where the $E_{int}$ predictions agree within their uncertainty estimates. Pink symbols denote dimers for which the predictions do not agree within uncertainty estimates, all of these feature an H-bond between the monomers.

With the aim of providing a systematic investigation of QM and MM approximations, we include a wide selection of methods. First, we study a variety of DFT functionals (e.g., global, range-separated, and double hybrids) with dispersion interactions selected from previous benchmark studies[67,68], namely PBE0+MBD, PBE0+D4, $\omega$B97X-V, $\omega$B97X+D3, $\omega$B97M-V, PBE+MBD, PBE0+XDM, PBE-QIDH+D3, CAM-B3LYP+XDM, B3LYP+D3, M06-2X, PBE0+MBD-NL, and PBE0+TS.

Second, among the SE methods we study the third-order Density Functional Tight Binding DFTB3+MBD[15] and GFN2-xTB[16]. From the available empirical classical FFs, we included GAFF2 (computed with AMBER)[11] and CHARMM-CGenFF[12] (computed with OpenMM)[69]. The results are presented in Supplementary Table S1 and as a spread of $E_{int}$ predictions obtained with these methods with respect to the LNO-CCSD(T) reference values, $\Delta E_{int}$, shown in Fig. 4a. They are presented in

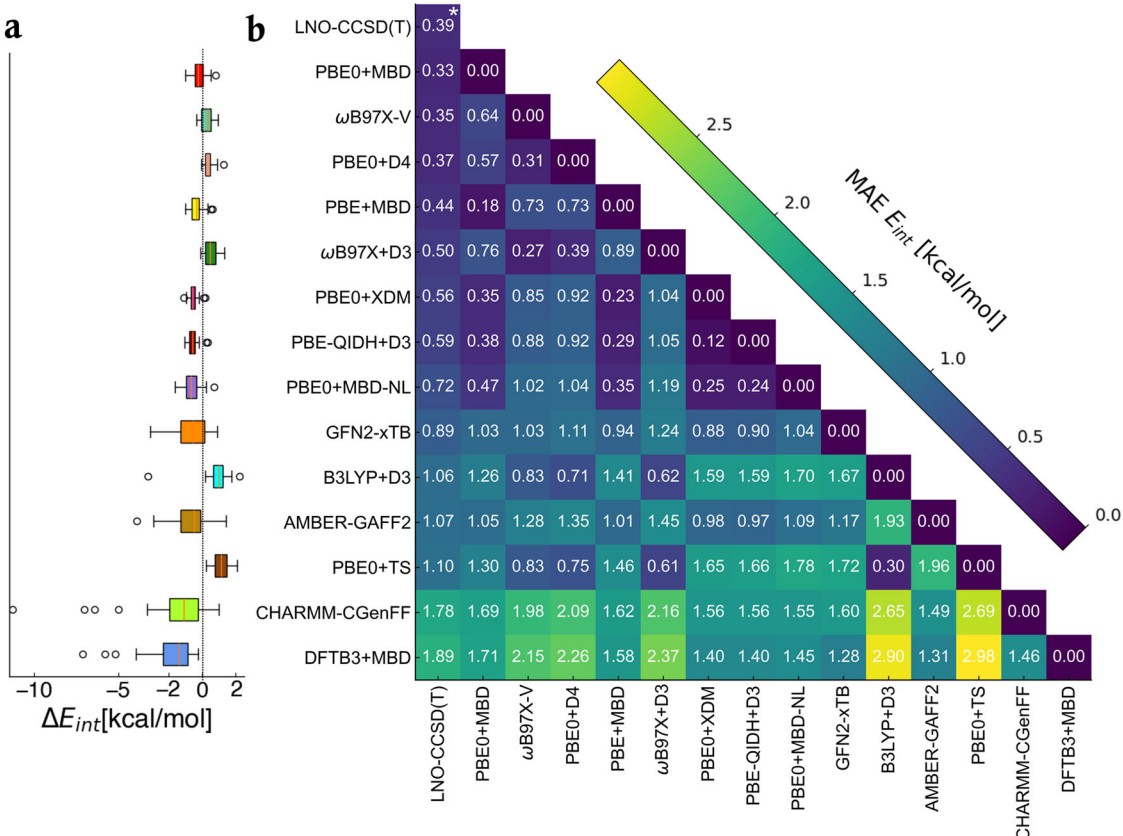

**Fig. 4 | Comparison of interaction energy predictions using high level and approximate methods.** Specifically, Density Functional Theory (DFT), semi-empirical, and empirical methods are compared to Local Natural Orbitals - Coupled Cluster with Singles, Doubles, and perturbative triplets (LNO-CCSD(T)[59–62]) and to each other. **a** Distributions of interaction energy $E_{int}$ predictions w.r.t. LNO-CCSD(T), $\Delta E_{int}$, showed via box plots, for a selection of computational methods - DFT methods: PBE0[17]+Many-Body Dispersion (MBD)[18,19], $\omega$B97X-V[70], PBE0[17]+D4[23,24], PBE[87]+MBD[18,19], $\omega$B97X[21]+D3[89,90], PBE0[17]+ eXchange-hole Dipole Moment (XDM)[95], PBE-QIDH[88]+D3[89,90], PBE0[17]+MBD-NL[20], B3LYP[91,92]+D3[89,90], PBE0[17]+TS[25]; semi-empirical methods: DFTB3[15]+MBD[18,19], GFN2-xTB[16]; and classical force fields: AMBER-GAFF2[11] and CHARMM-CGenFF[12]. The negative $\Delta E_{int}$ values signify underbinding, while the positive ones overbinding. Each boxplot shows the median of the error distribution with a vertical median, a box covering the interquartile range from the 25th to the 75th percentile is shown in color, with whiskers spanning horizontally 1.5 times the interquartile range and data points outside that range plotted individually as outliers. **b** A heatmap of mean absolute error (MAE) values of predicted $E_{int}$ w.r.t. LNO-CCSD(T) for the 42 QUID equilibrium dimers in the first column, and the MAE of all methods w.r.t. each other in subsequent columns. The computational methods to predict $E_{int}$ were the same methods as in (**a**). *For the LNO-CCSD(T) method, the value shown with asterisk is the mean absolute of the uncertainty estimates for $E_{int}$.

ascending order of MAE, whose values can be found in the first column of Fig. 4b (full results in Supplementary Fig. S7). The performance must be analyzed in the context of the intrinsic uncertainty estimate of the LNO-CCSD(T) method on the QUID dataset (mean value 0.39 kcal/mol). From this perspective, PBE0+MBD, $\omega$B97X-V, and PBE0+D4 are within the uncertainty of LNO-CCSD(T), although a case-by-case analysis reveals deviations beyond the uncertainty of the benchmark data.

Overall, it is reassuring that all recent DFT approximations yield rather accurate results. Even if some of them underestimate (PBE0+XDM, M06-2X, PBE0+MBD-NL, PBE-QIDH+D3) or overestimate (B3LYP+D3, PBE0+TS) the reference interaction energies on average, the spread of deviations is rather narrow for *all* DFT methods (with B3LYP+D3 being the only exception in the def2-QZVPPD basis set). On the other hand, both empirical and semiempirical methods show a tendency to underbind, producing larger spreads and exhibiting large outliers. The most prominent outliers are found for the methods CHARMM-CGenFF and DFTB3+MBD, with errors ranging from −12.5 kcal/mol to −5 kcal/mol (details in Supplementary Fig. S6) and −7.5 kcal/mol to −4.5 kcal/mol, correspondingly. Examining the DFTB3+MBD outliers—SF1I2, SF3I1, and SF3I3 dimers—reveals that for SF1I2 the strongest interactions are driven by electrostatics, including contributions from a sulfonyl group at the binding site, while the SF3Ix dimers exhibit reactive thiazole groups. It is noticeable that the error

distributions of GFN2-xTB (a SE method) and AMBER-GAFF2 (an empirical FF) are quite similar, although GFN2-xTB has a slightly lower average deviation from LNO-CCSD(T). Since GFN2-xTB was partially fitted to CCSD(T) data while AMBER-GAFF was not, this is not surprising.

For AMBER-GAFF2, dimers with high electrostatic contributions result in larger errors (see Supplementary Fig. S5) pointing to a limitation of fixed partial charges. Contrarily, for GFN2-xTB, the higher errors appear to be associated with the local chemical environment. For example, the presence of P (in all F2Ix dimers), S (in all SF3Ix dimers) or Cl atoms (in both Folded F1I1, F1I3 and Linear structures L1I2, L2I2) as well as H-bonds (in L3I3) affects the bonding of imidazole ligands, presenting greater challenges for the method.

To evaluate the level and areas of agreement between methods, we have also computed the MAE values for $E_{int}$ of QUID equilibrium dimers relative to each other (see Fig. 4b and Supplementary Fig. S6). The functionals $\omega$B97X-D3 and $\omega$B97X-V show excellent agreement with each other (MAE = 0.25 kcal/mol) despite the distinct incorporation of dispersion terms, D3 and non-local correlation VV10, in $\omega$B97X+D3 and $\omega$B97X-V, respectively[21,70]. This indicates that the critical similarity between $\omega$B97X-V and $\omega$B97X+D3, which sets them apart from the Minnesota functional M06-2X or PBE0+MBD, is the range-separation treatment of the DFT functional[71]. Particularly important for

QUID systems appears to be the long-range handling of the electron-electron interactions, as the short-range ones differ for the GGA and meta-GGA functionals[71]. In the same vein, we consider the related PBE0+MBD and PBE0+MBD-NL methods (MAE of 0.47 kcal/mol)—we notice that the MBD-NL method increases the deviation compared to MBD in almost all cases except for the dimer with two S atoms and imidazole ligand, SF3I1-3 (see Supplementary Fig. S6). We note that the MBD-NL functional was designed to achieve broad applicability to inorganic solids and molecular systems, while the original MBD (or MBD@rsSCS) method was developed for molecular systems. Interestingly, while the PBE0+MBD and the PBE+MBD functionals produce comparable results, with MAEs of 0.33 kcal/mol and 0.44 kcal/mol, respectively, the double-hybrid PBE-QIDH+D3 functional, based on the same PBE method, achieves an MAE of 0.59 kcal/mol (2.63 kcal/mol without the D3 correction). This may be a result of the shortcomings found in MP2 contributions for large flexible $\pi$–$\pi$ dispersion-dominated molecular systems[39] and the role of van der Waals parameterizations for double hybrid functionals[72].

Let us now investigate more in-depth the performance of the three best performing methods PBE0+MBD, $\omega$B97X-V, and PBE0+D4, all of which obtain $E_{int}$ within the LNO-CCSD(T) uncertainty estimate (0.39 kcal/mol) at 0.33 kcal/mol, 0.35 kcal/mol, and 0.37 kcal/mol, respectively. As the chemical environment and energetic balance in the NCIs proved to be a more distinguishing factor for the method than the structure types, we focus on a consideration of dispersion versus electrostatics contributions to $E_{int}$. Overall, the MAE value of the 14 electrostatics-dominated dimers for PBE0+MBD is 0.26 kcal/mol, notably better than the PBE0+D4 results with an MAE of 0.56 kcal/mol. On the other hand, for the 28 dispersion-dominated dimers, PBE0+D4 yields 0.27 kcal/mol, while PBE0+MBD obtains a close MAE of 0.35 kcal/mol. This suggests that systems with stronger electrostatic interactions pose a greater challenge for the D4 dispersion correction. This could stem from the different underlying mechanisms of the two approaches for modeling long-range correlation effects[18,19,24,73,74]. $\omega$B97X-V achieves correspondingly 0.25 kcal/mol, outperforming for the electrostatics-dominated dimers but has higher MAE of 0.40 kcal/mol for the dispersion-dominated ones.

In summary, empirical and semiempirical methods have demonstrated variable performance for ligand-pocket model systems in QUID, yielding a MAE of about 1 kcal/mol or higher and exhibiting a tendency to underbind. In contrast, among the many DFT methods examined, PBE0+MBD, $\omega$B97X-V, and PBE0+D4 proved most effective in capturing the complex QM effects contributing to $E_{int}$ calculations, while PBE0+XDM also showed excellent performance as a pairwise dispersion method. These findings enhance our understanding of the applicability and limitations of the various investigated computational methods. However, the choice of method for simulating a ligand binding to a protein pocket should also account for non-equilibrium conformations.

## Non-covalent bond dissociation pathways: non-equilibrium dimers

A key factor in modeling the dynamics of ligand-pocket systems is the capability of a physical model to investigate systems out of equilibrium accurately. To that end, we have considered six representative dissociation curves (i.e., F2B1, F2I1, L2B3, L2I3, SF2B2, and SF2I2) and conducted an in-depth analysis of the performance of selected computational methods: PBE0+MBD, PBE0+D4, PBE0+XDM, GFN2-xTB, DFTB3+MBD, and AMBER-GAFF2. The choice of DFT functional was based on the long-range effects in the elongated non-covalent bond regime as seen in Supplementary Table S2 and allowed for direct comparison within the same functional PBE0. In Fig. 5a, we present the averaged results (over six dimers) for the 'Delta metric' $\Delta$ that measures the agreement between the dissociation curves of a given computational method and the LNO-CCSD(T) reference

(see Supplementary Fig. S8). Indeed, it is evident that SE and classical FF methods, which are the tools of choice for biomolecular modeling, perform notably worse than DFT methods. The $\Delta$ values per dimer are listed in Supplementary Table S3, with the best performing equilibrium methods PBE0+MBD and PBE0+D4 achieving smaller $\Delta$ values. These findings are confirmed by analyzing the average error of $E_{int}$ w.r.t. LNO-CCSD(T) at each $q$, see Fig. 5b (corresponding six individual plots are available in Supplementary Figs. S9–S14). Notably, the performance of each method shows a strong dependence on the intermolecular distance. To elucidate the results, the dissociation curve profiles for all methods are presented individually in Supplementary Figs. S9–S14. Interestingly, unlike DFT methods, AMBER-GAFF2 either under-estimates or overestimates $E_{int}$, depending on the dimer configuration. The discrepancies are more pronounced in configurations where dispersion components dominate the NCI, and for those dimers particularly at distances with factor $q < 1.0$, where dispersion interactions are stronger. On the other hand, both SE methods predominantly underestimate $E_{int}$ and fail to accurately capture the position of the minimum on the dissociation curve or its overall shape. This behavior changes only for GFN2-xTB at $q < 1.0$, where it overestimates $E_{int}$ in most cases. Hence, to assess the efficiency of the methods in different interaction regimes, two ranges have been defined: 'compressed' for $q \le 1.0$ and 'elongated' for $q > 1.0$.

Concerning DFT methods, the best performance across the dissociation curves is consistently displayed by PBE0+MBD (underestimation) and PBE0+D4 (overestimation), with errors remaining within the uncertainty estimates of LNO-CCSD(T). The MAE values for $E_{int}$ in the 'compressed' and 'elongated' regimes obtained using all methods are provided in Supplementary Table S2. As expected from previous results, PBE0+D4, $\omega$B97X-V, $\omega$B97X+D3, PBE0+D4, and PBE0+MBD yield the best results in the 'compressed' regime, while PBE0+MBD, PBE+MBD, PBE0+XDM, PBE0+D4, $\omega$B97M-V, and $\omega$B97X-V perform best in the 'elongated' regime, indicating the consistently good performance of the PBE0 and $\omega$B97X functionals. The SF2B2 and SF2I2 dimers proved to be the most challenging among those examined, likely due to the interaction of a 5-membered oxadiazole ring ($C_2N_2O$) via $\pi$–$\pi$ stacking with the small monomer. The presence of two N and one O atoms in the aromatic ring contributes to an increase in the dipole moment and polarizability of the monomer, thereby enhancing both electrostatic and dispersion interactions. Interestingly, F2B1 and F2I1 are the easiest dimers to predict among the examined methods, as the molecular environment contributing to NCI is located within a few Å of the molecule.

While the analysis of the dissociation curves has confirmed the performance of the methods for computing $E_{int}$ (*vide supra*), it has also revealed that the accuracy of SE methods and classical FF strongly depends on the distance range and the dimer configuration. This is a critical result, as both approaches are widely used to investigate intermolecular interactions in biomolecular simulations, raising questions about the reliability of the results obtained in molecular dynamics simulations carried out with empirical methods.

## Quantum-mechanical property space of QUID systems

To enhance our understanding of the effects of dimer configuration and intermolecular distances on the properties of pocket-ligand systems, several global and local physicochemical properties, in addition to $E_{int}$, were computed for all equilibrium and non-equilibrium QUID dimers at PBE0+MBD level of theory (see "Methods" section for more details). A full list of properties, similar to those in the Aquamarine[30] dataset of large monomers, is provided in Supplementary Table S5. Further, quantities describing the Hirshfeld partitioning, i.e., Hirshfeld volumes, ratios, charges, (scalar) dipole moments provides information for the electron response of an atom-in-molecule environment. Overall the molecular properties can also be

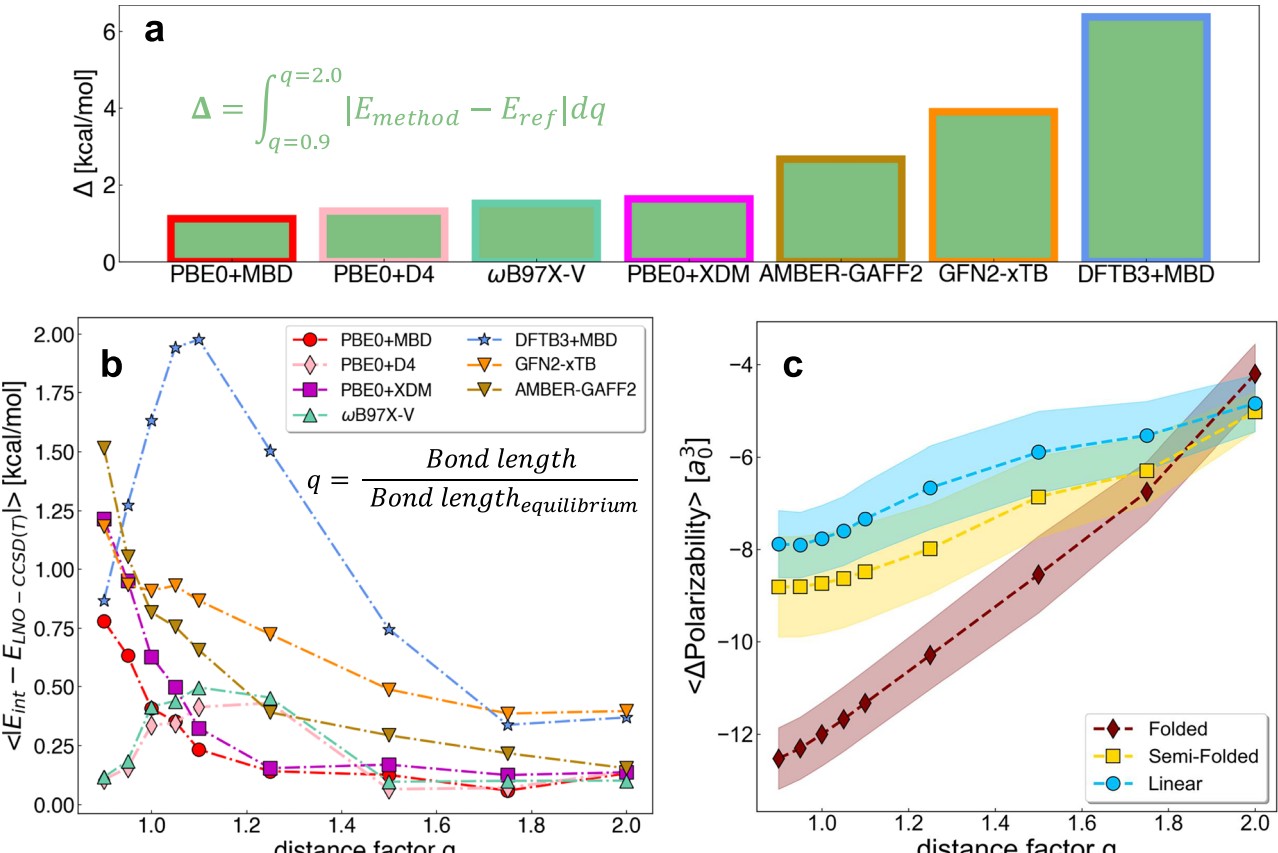

**Fig. 5 | Interaction energy and molecular polarizability along non-covalent bond dissociations. a** The delta metric ($\Delta$, see formula on the plot) results are shown for four Density Functional Theory methods: PBE0[17] including Many-Body Dispersion (MBD)[18,19], D4[23,24], and eXchange-hole Dipole Moment (XDM)[95], $\omega$B97X-V[70]; two semiempirical methods: DFTB3[15]+MBD[18,19], GFN2-xTB[16]; and a classical force field method: AMBER-GAFF2[11]. **b** Average of the absolute difference of predicted interaction energy with the Local Natural Orbitals - Coupled Cluster with Singles, Doubles, and perturbative triplets (LNO-CCSD(T)) reference along the dissociation of the non-covalent bond of a selection of dimers. The average is calculated at each multiplicative distance factor $q$ (ranging from 0.9 to 2.0), defined

as the ratio between the bond length and the equilibrium non-covalent bond length for the corresponding dimer. The average is shown for six selected molecular dimers: F2B1, F2I1, SF2B2, SF2I2, L2B3, and L2I3, using the same methods and corresponding color highlights as in (**a**). **c** Average of the difference in the molecular polarizability of the dimer and the sum of isotropic polarizabilities of its corresponding monomers at each distance factor $q$. The results are shown as an average (points) with a standard error (background) over all 128 non-equilibrium dimers in the QUantum Interacting Dimer (QUID) dataset, split by structural type in Linear, Semi-Folded, and Folded.

useful as ML descriptors[75]. Here, we first focus on the molecular polarizability, $\alpha$, as an additional measure of the NCIs.

To that end, analogous to $E_{int}$, the difference in $\alpha$ between each dimer and the sum of its corresponding large and small monomers, $\Delta\alpha$, was calculated for all 128 non-equilibrium conformations. The average $\Delta\alpha$ values for each structure type as a function of the distance factor $q$ are plotted in Fig. 5c). Overall, the three structure types exhibit an almost linear behavior, with slight deviations near the equilibrium distance for the Linear and Semi-Folded structures. According to a recent concept of chemical bonding based on $\alpha$, proposed by D. Hait and M. Head-Gordon[76], this linear behavior suggests no significant modification in the covalent bond arrangements along the dissociation curve. The variation can thus be attributed to the self-consistent screening effect between the monomers. Indeed, in both Linear and Semi-Folded structures, the small monomer affects fewer atoms of the large monomer. In contrast, in Folded structures, the small monomer remains within 5 Å of a significant number of atoms of the large monomer, substantially influencing the electrostatics and dispersion in the pocket site, resulting in a steeper curve. Moreover, no correlation between $\alpha$ with $E_{int}$ and $\mu$ emerges from the exploration QUID dimers (see Supplementary Fig. S15). On the one hand, this could suggest flexibility for rational ligand design as observed for small molecules of up to 7 heavy atoms[75]. At this stage, we can ascertain the

interplay between electrostatic and dispersion interactions in a structurally and chemically complex local environment in the QUID pocket-ligand proxies requires exact QM models beyond the capture of a single key global property in the high-dimensional chemical compound space.

Another relevant property for understanding NCIs in QUID dimers is the atomic forces, which are widely used to parameterize MLFFs for (bio)molecular systems. Indeed, the molecular conformational sampling at a given temperature strongly depends on the accuracy of the chosen computational method in adequately describing the forces acting on the atoms in the molecule. Accordingly, we have analyzed the atomic force distributions using a selection of DFT methods: the best performing ones PBE0+MBD and PBE0+D4, and the well-performing pairwise PBE0+XDM method. Since these methods share the DFT functional and our primary interest lies in NCIs, the focus will be on the vdW components of the atomic forces. We take the MBD method as the reference for comparison because geometry optimizations have been carried out with the PBE0+MBD level of theory. In addition, MBD is the only method that includes non-local screening effects in the polarizability, which were shown to be important in Fig. 5c).

The differences in the vdW atomic forces are examined in terms of their magnitude and direction, treated as two distinct yet interrelated driving factors in MD simulations. The results are presented in Fig. 6a as a

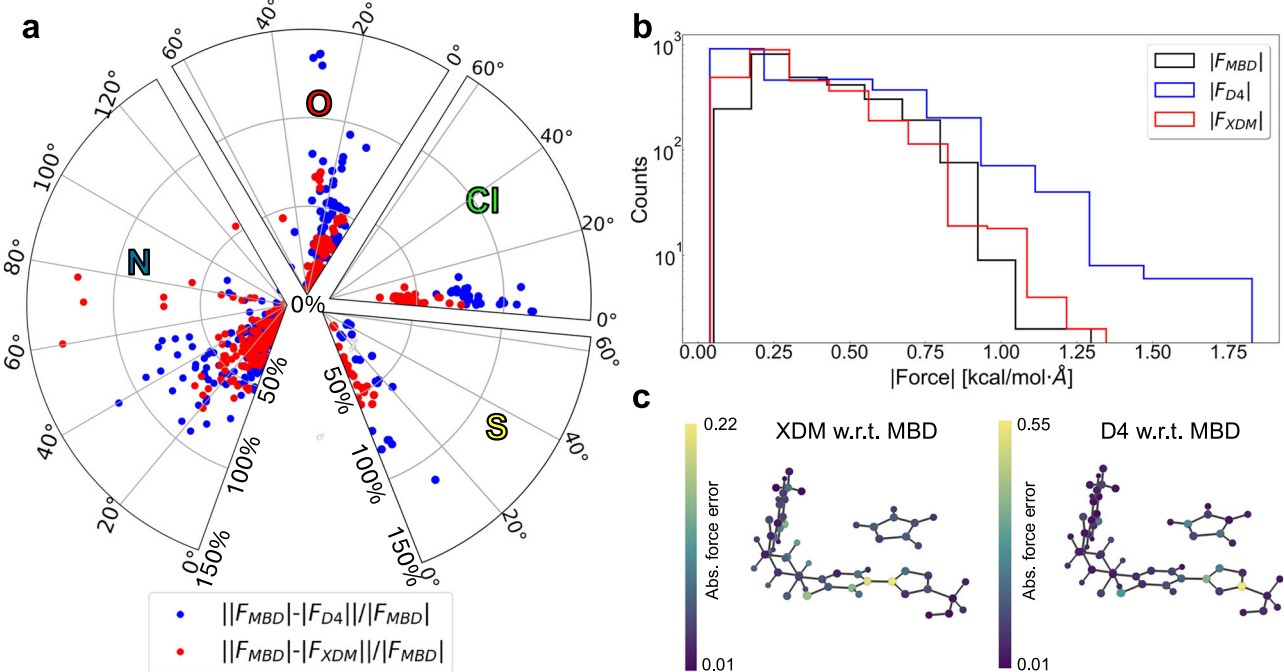

**Fig. 6 | Atomic van der Waals (vdW) force differences between Many-Body Dispersion (MBD)[18,19], D4[23,24], and eXchange-hole Dipole Moment (XDM) methods[95]. a** A polar plot, where the radius is the difference in magnitude of the vdW forces (given as a percentage) and the angle is the arccosine between the dot product of the force vectors. Four such polar plots depict the distributions of the forces acting on all the N, O, Cl, and S atoms, respectively, in all 42 equilibrium

QUantum Interacting Dimer (QUID) structures. **b** Overall distribution of the magnitudes of the atomic vdW forces on all atoms for all 42 equilibrium QUID dimers using the MBD, D4, and XDM methods. **c** An example of the deviation in atomic van der Waals (vdW) forces between different methods is visualized for the SF2I2 dimer (produced by the FFAST software[77]). The range is represented by colors varying from purple to yellow in a viridis color bar.

radial plot illustrating the difference in forces of D4 and XDM with respect to MBD. In this plot, the angular part represents the arccosine between the dot product of two force vectors, while the radius is the difference in force magnitudes, scaled by the MBD force. This analysis focuses on the differences observed per atom type, with the results for N, O, Cl, and S atoms shown in Fig. 6a (see other atom types in Supplementary Fig. S16).

The overall force analysis is provided by the plot in Fig. 6b, which shows that MBD yields the smallest forces on average. The vdW force magnitudes are generally not much higher than 1 kcal/mol/Å; this is expected given that equilibrium geometries are involved. Higher vdW forces would be expected for non-equilibrium geometries. Further, largest discrepancies in force magnitude were found in D4 compared to XDM. This is corroborated by the distribution of the atomic force magnitudes for all atoms. The most significant outliers in force magnitude discrepancies are associated with D4 and Fig. 6a demonstrates that those outliers of up to 3–4 times the magnitude of the MBD force are found on C atoms in descending order for L2I3, L2B3, L3I1, and L3I2 (Supplementary Fig. S16). The FFAST software[77] allows for visualization of the discrepancies between the vdW atomic forces to confirm that the C atoms were found at the binding site, an example of such a visualization can be seen in Fig. 6c) depicting SF2I2. The difference in force magnitudes is visible not only on the atoms of the binding site as expected but also in proximity to it showing the differing impact of the ligand interactions in the different methods. For the SF2I2, there is a notable difference between the comparisons of XDM and D4 w.r.t. MBD (the MBD depiction is available in Supplementary Fig. S17).

This also holds true in general for the vdW forces on the Cl atoms in the QUID dimers. As seen on Fig. 6a XDM is in better agreement with MBD than D4 for Cl atoms, and the same is true for the other halogen element F, as well as the few P atoms (see Supplementary Fig. S16). Hence, the presence of more electronegative atoms can hint at a systematic difference between the different vdW atomic forces. In that vein, a particular outlier is seen for the S element, with a 113% gap

between the magnitude of the MBD and D4 forces for the SF1I1 dimer, in the sulfonyl group of the binding site of the imidazole ligand (see Fig. S4). Interestingly, there is a split in the vdW force directions between 'heavier' atoms in the QUID dimers, i.e., O, F, Cl, S, and P and the 'lighter' ones i.e., H, C, and N. The 'lighter' atoms represented more in organic molecules demonstrate a larger spread of angle difference between the forces up to 120°–180°. By construction, for the pairwise D4 and XDM methods, the vdW force is a simple vector sum where all force vectors are aligned along the vector connecting pairs of atoms. In contrast, many-body effects that are treated to infinite order in MBD can thus substantially alter the force directions, and this difference is much more pronounced than for force magnitudes. This could have a visible effect on MD trajectories, although these implications remain to be assessed in a future study.

The observed difference in force directions has potential repercussions for MLFF methods, where molecular data is routinely optimized at one level of theory, method or functional or dispersion correction, and computed at a different one to serve as input or reference for energies and forces. Unfortunately, while for $E_{int}$ we have highly accurate LNO-CCSD(T) reference data and even the ability to achieve a "platinum standard" confidence by comparing with FN-DMC, obtaining forces at benchmark ab initio level is prohibitively expensive. Current research in developing gradients for LNO-CCSD(T)[78] and FN-DMC[79] could facilitate a reference benchmark for the vdW forces in the future and provide a clearer picture for the accuracy of the DFT methods for MD of ligands binding to a pocket.

## Discussion

The "Quantum Interacting Dimer" (QUID) benchmarking framework was developed here to redefine the state of the art in QM-based modeling of ligand-pocket motifs. Currently, QUID contains 170 structurally and chemically diverse large molecular dimers (42 equilibrium and 128 non-equilibrium) of up to 64 atoms, including the

H, N, C, O, F, P, S, and Cl chemical elements, encompassing most atom types of interest for drug discovery purposes. This diversity enables a single dimer to exhibit multiple types of steric effects and NCIs simultaneously, including, but not limited to, $\pi-\pi$ stacking, hydrogen, and halogen bonds. Accordingly, we conducted a series of analyses that provided valuable insights into inter- and intramolecular interactions of these model protein-ligand systems from a QM perspective.

Indeed, the energy decomposition of the interaction energy $E_{int}$, as obtained through SAPT analysis, revealed that ligand-pocket interactions are predominantly governed by dispersion and electrostatics—types of interactions often inadequately represented by MM methods. Moreover, we defined a "platinum standard" of accuracy for $E_{int}$ of ligand-pocket interactions by contrasting the results with the "gold standard" methods such as LNO-CCSD(T) and FN-DMC. Notably, the previously reported disagreement between LNO-CCSD(T) and FN-DMC for large non-covalent systems[56] was not observed at such scale for the QUID dimers, where the disagreements are driven by predominantly electrostatic interactions as in a recent study on the S66 dataset[57], and the overall discrepancy is small, approximately 0.5 kcal/mol. Those identified dimers can serve as model systems for future exhaustive analysis in both QM methods. Thus, our findings demonstrated that among all studied MM and QM approaches, DFT methods such as PBE0+MBD, $\omega$B97X-V, and PBE0+D4 achieve excellent agreement with the costly LNO-CCSD(T) method in the determination of $E_{int}$ for equilibrium and non-equilibrium dimers. Additionally, we have identified certain limitations in widely used semiempirical (e.g., GFN2-xTB and DFTB3+MBD) and MM methods (e.g., AMBER-GAFF2 and CHARMM-CGenFF) for investigating complex non-covalent motifs, which raises questions about their reliability in binding affinity simulations. These intriguing results highlight the relevance of determining the appropriate level of theory to accurately characterize protein-ligand systems, particularly in the development of extensive QM datasets utilized in physical method benchmarking and ML-based investigations.

Furthermore, QUID provides access to a diverse set of extensive and intensive (global and local) QM properties beyond $E_{int}$ (at PBE0+MBD level), enabling the electronic characterization of chemical environments within ligand-pocket motifs—a limitation of current benchmark frameworks, which primarily focus on structural and energetic features. The structural dependence of the polarizability change in dimers as a function of the monomer separation, as well as the lack of correlation between global electronic properties and $E_{int}$, offers a perspective for understanding the NCIs in protein-ligand systems. These insights can rationalize the design of drug-like molecules targeting specific pocket sites with a desired set of QM properties[75]. The observed discrepancies in the van der Waals (vdW) component of the atomic forces using MBD, D4, and XDM methods also show the importance of investigating additional properties beyond the traditional $E_{int}$ in non-covalent complexes. An inaccurate description of vdW forces can strongly impact the reaction pathway and the resulting binding pose when simulating the interaction of ligands with protein pockets. Hence, QUID has the potential to revolutionize standard procedures in approaching the modeling of ligand-pocket interactions in physical and ML-based frameworks by providing global and local electronic property data for large molecular dimers and their building blocks, which are critical for a faithful incorporation of long-range effects[46,47].

In summary, the QUID benchmark framework presents a rigorously designed approach for accurately analyzing ligand interactions at various protein binding sites, facilitating the development of robust QM datasets for reliable predictions of binding affinities and structural conformations. While the insights gained from this work highlight the importance of an appropriate QM description for inter- and intramolecular properties of ligand-pocket motifs, we acknowledge that further efforts should incorporate more flexible and charged pocket structures, as well as solvation effects[30,80]. The sampled chemical space should ultimately encompass full pocket-ligand molecular systems, similar to those in MM datasets (e.g., PL-REX[81] and QR50[82]), to enhance the reliability of the findings. We expect this work to pave the way for a more informed use and refinement of physical and chemical models for simulating ligand-pocket interactions, offering particular value in fine-tuning MLFFs and ML-augmented semiempirical models, which are increasingly integrated into screening pipelines for drug discovery.

## Methods

### Generation procedure of QUID systems

A schematic representation of the procedure used to generate the 170 (equilibrium and non-equilibrium) QUID dimers is outlined in Fig. 1. Each QUID dimer consists of one of nine chemically diverse large monomers (of ≈50 atoms) selected from the Aquamarine dataset of drug-like molecules[30], paired with a small monomer, either benzene $C_6H_6$ or imidazole $C_3H_4N_2$. Each large monomer features at least 2 sterically accessible aromatic rings, which serve as binding sites. The larger monomer initially has chain-like geometry, modeling a protein pocket, while the small molecule represents the ligand. The chemical composition of QUID systems reflects this purpose: each conformer contains not only C, N, O, and H atoms but also at least one of the elements F, P, S, or Cl for eight out of the nine large monomers. These elements are incorporated into the following functional groups: substituted five- or six-membered aromatic rings, aliphatic heterocycles, ketones, ethers, hydroxyls, amines, haloalkanes, and sulfonyl. Moreover, the choice of benzene and imidazole molecules enables a comparison of the effects of small aromatic rings and amphoteric compounds (imidazole can be both H-bond donor and acceptor), which is a key characteristic for many compounds of biomedical significance such as amino-acid histidine and anti-inflammatory drugs[48].

In the initial dimer conformation, prior to structural optimization, benzene and imidazole occupy the same position relative to a given binding site, specifically at 3.55 ± 0.05 Å parallel to the site. A similar distance was used in the generation procedure of S66x8 dataset[38]. Here, the aromatic ring of the small monomer was aligned with that of the binding site, ensuring that the distance between the corresponding pair of heavy atoms on the two rings remained within a margin of 0.05 Å. Each dimer was then optimized using non-empirical hybrid density functional theory (DFT) with many-body dispersion approach (range-separated self-consistent screening, MBD@rsSCS), namely PBE0+MBD, in conjunction with tightly-converged numeric atom-centered orbitals ("tight settings"), as implemented in the `FHI-aims`[83] software, version 221103. We classify the resulting structures into three categories based on their shape: 'Linear', where the original chain-like geometry is mainly preserved; 'Semi-Folded', where parts of the large monomer bend while other sections remain linear; and 'Folded', where the large monomer encases the smaller ones. Following this classification, the first letter of the dimer names corresponds to the structural category: 'F' for Folded, 'SF' for Semi-Folded, and 'L' for Linear. The small monomer is indicated by the letter 'B' for benzene or 'I' for imidazole, while the number that follows indicates the binding site (in QUID dimers as found viable post-optimization). The chemical diversity of the equilibrium QUID dimers (represented by the counts of non-H elements) is plotted in Supplementary Fig. S1 against the radius of gyration, illustrating the spread of the three structural categories. The software Avogadro[84] (version 1.2.0) and Visual Molecular Dynamics (VMD)[85] (version 1.9.4a57-arm64-Rev12) were used for visualization of the molecules.

Furthermore, we acknowledge that modeling ligand-pocket systems would be incomplete without computing non-equilibrium dimer conformations, which are crucial for understanding the binding of drugs to protein pockets. To this end, we generated eight optimized out-of-equilibrium structures along the reaction pathway of the non-covalent bond for 16 representative molecular dimers (F2B1, F2I1, F2B2, F2I2,

SF2B1, SF2I1, SF2B2, SF2I2, SF2B3, SF2I3, L2B1, L2I1, L2B2, L2I2, L2B3, L2I3). These non-equilibrium conformations were constructed with inter-monomer distances ranging from 3.2 to 7.1 Å, defined by a dimensionless multiplicative factor $q$, which represents the ratio of the inter-monomer distance in a given conformation to that in its equilibrium state. The chosen values of $q$ are 0.90, 0.95, 1.00, 1.05, 1.10, 1.25, 1.50, 1.75, and 2.00, as shown in Fig. 1. For the $\pi-\pi$ interactions between aromatic rings, the dissociation vector was defined as the distance between the plane of the aromatic ring on the large monomer and the center of mass of the small monomer. In the case of H-bonds, dissociation was measured between the proton acceptor and donor atoms. All the non-equilibrium geometries were also optimized at PBE0+MBD level with "tight" settings using FHI-aims (version 221103). Unlike the equilibrium dimers, the heavy atoms of the binding site and small monomer were kept frozen in their respective positions during optimization. An example of the dissociation geometries for one of the non-equilibrium dimers is shown in Fig. 1.

### Property calculation

The interaction energies $E_{int}$ of QUID dimers were calculated using the supramolecular approach,

$$E_{\text{int}} = E_{dimer} - (E_{L_{monomer}} + E_{S_{monomer}}). \tag{1}$$

Counterpoise corrections were applied to PBE0+MBD, PBE-QIDH+D3, and CCSD(T) single-point calculations. The basis set superposition error was negligibly small (under 1.5%) for DFT and ca. 4% on the average for CCSD(T) when extrapolated to the complete basis set (CBS) limit (see results in Supplementary Fig. S18). To investigate the level of agreement among QM methods for calculating $E_{int}$ of QUID dimers, we have considered a selection of well-performing hybrid, double hybrid, range-separated hybrids DFT functionals, including M06-2X[22], $\omega$B97X+D3[21], $\omega$B97M-V[86]$\omega$B97X-V[70], PBE+MBD[87], PBE-QIDH+D3[88–90], B3LYP+D3[89–92], CAM-B3LYP+XDM[93] and BH&HLYP+XDM[94]. Additionally, the PBE0 functional was combined with multiple two-body or many-body corrections: MBD[18,19] (range-separated self-consistent screening (MBD@rsSCS) approach), MBD-NL[20] (Non-Local), XDM[95] (eXchange Dipole Moment), TS (Tkatchenko-Scheffler)-vdW[25], D4[23,24], $\omega$B97X+D3[21], $\omega$B97M-V[86]. These calculations were performed using either the FHI-aims (version 221103) software[96] with "tight" settings, the Psi4 software[97,98] (version 1.9.1) with the quadruple-zeta def2-QZVPPD basis set or the QChem software[99] (version 6.1) with the quadruple-zeta def2-QZVPPD basis set except in the CAM-B3LYP and BH&HLYP cases using the XDM dispersion, where the aug-cc-pVTZ basis set was used in accordance with previous studies[68] and the available parametrization from literature[100]. The PBE-QIDH-D3 implementation made use of the counterpoise correction as needed for the double hybrid functional method using MP2 correlation in the calculation[88]. All DFT calculations on Gaussian basis set employed the resolution of identity RI technique to accelerate the calculation of electron repulsion integrals. Notably, for the XDM method, a specific parametrization for the PBE0 functional was applied and then computed using FHI-aims software on "tight" settings ($a_1 = 0.4710$ and $a_2 = 2.3857$)[101]. SAPT energy decomposition calculations were carried out at the sSAPT0/jaDZ level[52] employing the Psi4 software[97] (version 1.9.1). At the semiempirical level, $E_{int}$ was calculated via single-point calculations using Density Functional Tight Binding[15] DFTB3+MBD with DFTB+ software[102] (version 23.1) and GFN2-xTB with the xTB software[16] (version 6.7.0).

Regarding MM methods, the results for AMBER[11] were obtained using Openbabel[103] for molecular format conversion. The parametrization with AmberTools (version 23.6) and GAFF2[11] required manual assignment and adjustment of bonds for more complex cases, such as ring interactions, as well as modification of the self-consistent loop limits for the F2I1 dimer. The CHARMM-CGenFF[12] calculations were conducted using OpenMM[69] (version 8.1.2) following a

CGenFF2[104] parametrization. For these calculations, manual inclusion of the dihedral angles for the flexible side chains, such as the 'C-C-N' type, was necessary. An example is the L4B1 dimer, which was assumed to exhibit relatively low flexibility due to the nature of its bonds and chemical environment. A list of the $E_{int}$ storage details is available in Supplementary Table S4.

Additionally, the optimized structures of equilibrium and non-equilibrium QUID dimers were also utilized for more accurate QM single-point calculations using PBE0+MBD level of theory to compute other physicochemical properties (as detailed in Supplementary Table S5). For these calculations, we have used the FHI-aims code[96] together with "tight" settings for basis functions and integration grids. Energies were converged to $10^{-6}$ eV and the accuracy of the forces was set to $10^{-4}$ eV/Å. The convergence criteria used during self-consistent field (SCF) optimizations were $10^{-3}$ eV for the sum of eigenvalues and $10^{-6}$ electrons/Å$^3$ for the charge density. The MBD energies and MBD atomic forces were here computed using the range-separated self-consistent screening (rsSCS) approach[19], while the atomic $C_6$ coefficients, isotropic atomic polarizabilities, molecular $C_6$ coefficients and molecular polarizabilities (both isotropic and tensor) were obtained via the SCS approach[18]. Here, we have also computed van der Waals forces using D4 and XDM methods. Hirshfeld ratios correspond to the Hirshfeld volumes divided by the free atom volumes. The TS dispersion energy refers to the pairwise Tkatchenko-Scheffler (TS) dispersion energy in conjunction with the PBE0 functional[25]. The vdW radii were also obtained using the SCS approach via $R_{vdW} = (\alpha^{SCS}/\alpha^{TS})^{1/3} R_{vdW}^{TS}$, where $\alpha^{TS}$ and $R_{vdW}^{TS}$ are the atomic polarizability and vdW radius computed according to the TS scheme, respectively. Atomization energies were obtained by subtracting the atomic PBE0 energies from the PBE0 total energy of each molecular conformation.

### LNO-CCSD(T) reference interaction energies

The large CCSD(T) computations well beyond the limits of conventional implementations were performed with the highly optimized local natural orbital (LNO) CCSD(T)[59–62] method in the MRCC[105,106] (version 2024) quantum chemistry suite. First, a detailed basis set and LNO approximation convergence study is performed to determine the most efficient LNO and basis set settings that provide high accuracy for the interactions relevant to QUID. To that end, the interaction energies were tested for three representative dimers, namely SF2I2 (Fig. 7), F2B1, and L2B3 (Supplementary Fig. S19), at 1×, 0.9×, and 2.0× equilibrium distances. Here, we used the systematically improving series of aug-cc-pV(X+d)Z basis sets with X = D, T and Q[107] as well as Normal (N), Tight (T), and very Tight (vT) LNO thresholds. The convergence toward the complete basis set (CBS) limit was accelerated for the HF[108] and correlation[109] energies via CBS extrapolation [CBS($X,X+1$)]. To accelerate the convergence toward conventional CCSD(T), that is the local approximation free (LAF) limit, the LAF extrapolations[61,62] N–T employs the Normal and Tight and the T–vT use the Tight and very Tight LNO settings.

Our best estimate of CCSD(T)/CBS for the above nine dimers is given by a composite ($\bar{E}$) scheme benefiting from CBS(T,Q) and LAF T–vT extrapolations as well as counterpoise corrections[110]:

$$E_{CCSD(T)}^{CBS} \approx \bar{E}_{T-vT\,LNO-CCSD(T)}^{CBS(T,Q)} \approx E_{T-vT\,LNO-CCSD(T)}^{CBS(D,T)} - E_{Tight\,LNO-CCSD(T)}^{CBS(D,T)} + E_{Tight\,LNO-CCSD(T)}^{CBS(T,Q)} \tag{2}$$

Moreover, an uncertainty estimate can be assigned to the LNO approximation as the difference between the usually monotonically converging steps, that is, $\pm 0.5|E_{veryTight}^{XZ} - E_{Tight}^{XZ}|$ for T–vT.

Inspecting all nine cases (Fig. 7 and Fig. S19), we find fast convergence with the LNO settings reaching ca. $\pm 0.1$ kcal/mol uncertainties at the T–vT LAF extrapolated level. Importantly, the N–T LAF extrapolated results are very close to the T–vT extrapolated ones, thus the most costly very Tight settings can be spared when targeting the

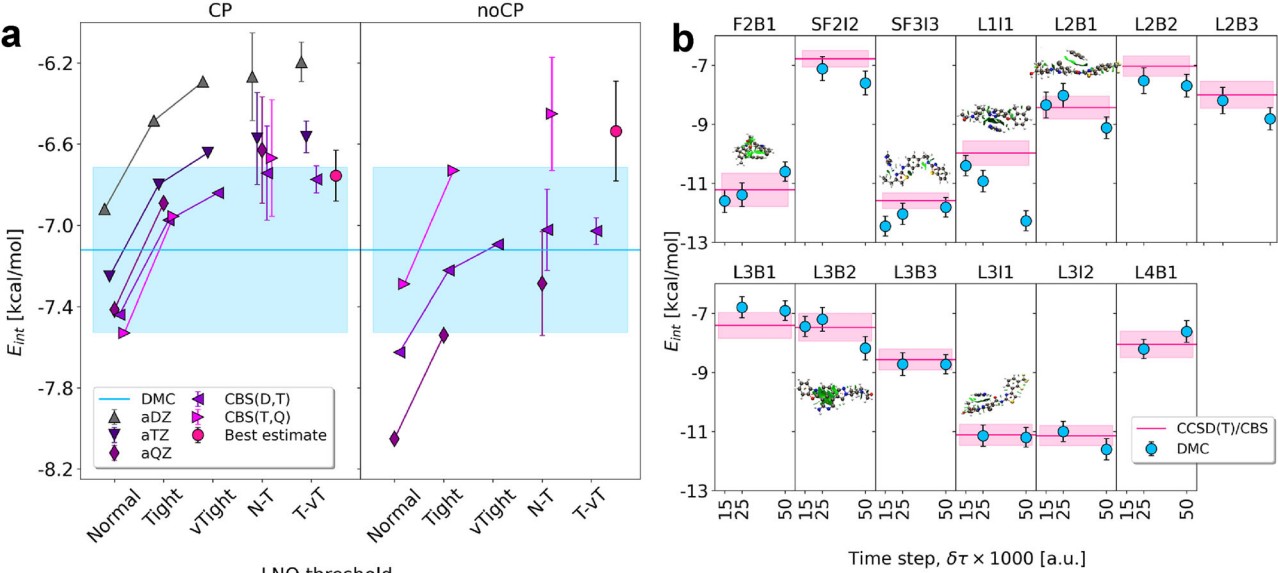

**Fig. 7 | Mutual agreement for interaction energies from benchmark ab initio methods LNO-CCSD(T) and FNDMC. a** Local Natural Orbitals - Coupled Cluster with Singles, Doubles, and perturbative triplets (LNO-CCSD(T)) interaction energy $E_{\text{int}}$ convergence analysis with respect to the LNO thresholds (x axis) and aug-cc-pV($X$+d)Z (aXZ) basis set choices with (CP) and without (noCP) CounterPoise corrections for the SF2I2 dimer at equilibrium distance, including the best estimate interaction energy corresponding to Eq. (2); the horizontal line indicates the Fixed-Node Diffusion Monte Carlo (FN-DMC) interaction energy (0.025 $\delta\tau$) with its

statistical error in a box. For FN-DMC, error bars represent estimated one-$\sigma$ statistical error for $4 \times 10^8$ configurations. For LNO-CCSD(T), the error bars correspond to the estimated uncertainty in the different extrapolations to the Complete Basis Set (CBS) and Local Approximation-Free (LAF) limits. **b** FN-DMC time-step convergence plots for a selection of 13 dimers. The reference LNO-CCSD(T) values are added for comparison, represented as horizontal lines with their uncertainty estimates shown as boxes.

entire dataset. Regarding the basis set convergence, for all nine dimers, we find a good agreement between the counterpoise corrected and uncorrected results by reaching the CBS(T,Q) level. An additional indication of excellent convergence is the close match between counterpoise corrected aug-cc-pV(Q+d)Z as well as CBS(D,T) results, also suggesting that the most expensive aug-cc-pV(Q+d)Z computations are not needed for the entire dataset. Furthermore, the difference of the best-converged basis set levels, e.g., between counterpoise corrected CBS(D,T) and CBS(T,Q), can be used as an uncertainty estimate for the basis set convergence. Then, our conservative formula for the total uncertainty estimate w.r.t. CCSD(T)/CBS adds together the above LNO and basis set uncertainty estimates, which indicates about ± 0.2 kcal/mol uncertainty at the $\bar{E}^{\text{CBS(T,Q)}}_{\text{T–vT LNO–CCSD(T)}}$ level [Eq. (2)].

On the basis of the above analysis, a more affordable composite energy expression can be recommended:

$$\bar{E}^{\text{CBS(D,T)}}_{\text{N–T LNO–CCSD(T)}} \approx E^{\text{DZ}}_{\text{N–T LNO–CCSD(T)}} - E^{\text{DZ}}_{\text{Normal LNO–CCSD(T)}} + E^{\text{CBS(D,T)}}_{\text{Normal LNO–CCSD(T)}}, \quad (3)$$

which will be used to compute reference $E_{\text{int}}$ values for QUID dimers. In accordance with the above analysis of the nine convergence plots, the more efficient $\bar{E}^{\text{CBS(D,T)}}_{\text{N–T LNO–CCSD(T)}}$ results are in excellent agreement with the tightly converged $\bar{E}^{\text{CBS(T,Q)}}_{\text{T–vT LNO–CCSD(T)}}$ ones, their difference is mostly below 0.1 kcal/mol, and only up to 0.2–0.3 kcal/mol for the worst cases of L2B3 (equilibrium) and SF2I2 (0.9× equilibrium). The corresponding uncertainty estimate for $\bar{E}^{\text{CBS(D,T)}}_{\text{N–T LNO–CCSD(T)}}$ is the sum of $0.5|E^{\text{DZ}}_{\text{Tight}} - E^{\text{DZ}}_{\text{Normal}}|$ and $0.33|E^{\text{CBS(D,T)}}_{\text{Normal}} - E^{\text{TZ}}_{\text{Normal}}|$, resulting on the average about ±0.4 kcal/mol, in agreement with the observations for the nine dimers where $\bar{E}^{\text{CBS(T,Q)}}_{\text{T–vT LNO–CCSD(T)}}$ is available.

The benefit of $\bar{E}^{\text{CBS(D,T)}}_{\text{N–T LNO–CCSD(T)}}$ over $\bar{E}^{\text{CBS(T,Q)}}_{\text{T–vT LNO–CCSD(T)}}$ is about an order of magnitude computer time and 4-fold memory demand

reduction. Namely, the three kinds of LNO-CCSD(T) interaction energy computations needed for Eq. (3) can be completed in ca. 30–70 hours for each dimer on 16 cores and at most 20 GB memory, making it ideal for high-throughput computations even on clusters with short walltime limits. For the sake of completeness, the frozen-core approximation and conventional auxiliary basis sets[111,112] were employed for all LNO-CCSD(T) interaction energies.

## FN-DMC interaction energy calculation

We performed Fixed-Node Diffusion Monte Carlo (FN-DMC) calculations[27–29] for QUID dimers corroborating the LNO-CCSD(T) $E_{\text{int}}$ with another accurate QM method. Our FN-DMC wave function ansatz was built using a single Slater determinant in addition to a Jastrow factor with one-body, two-body, and 3/4-body terms to account for cusps conditions at the nuclei, fermionic pair correlations, and product of pair correlations at the field of the nuclei, respectively. See references[113,114] for further details about the electronic wave function ansatz. The molecular orbitals of the Slater determinant were taken from a previous DFT calculation employing ORCA code[115] (version 5.0.4) with the Local-density approximation (LDA) exchange-correlation functional, a cc-pVTZ basis set for all atoms, and the ccECP pseudopotential[116].

All variational parameters of the Jastrow factor were variationally optimized at the Variational Monte Carlo (VMC) level with the stochastic reconfiguration optimization method[117], while molecular orbital coefficients, basis set contraction coefficients, and exponents were kept fixed from the initial DFT calculation. The optimized VMC wave functions were taken as guiding functions in the FN-DMC calculation, in which we also employed the ccECP pseudopotentials to approximate core electrons for each atom, integrated with the determinant localization approximation (DLA)[118]. Consequently, the FN-DMC calculations were computed at two time-steps of 0.050 and 0.025 (a.u.), using 12,800 walkers divided into 300 blocks, each 100 steps long, for a total of $4 \times 10^8$ sampled configurations. FN-DMC statistical error bars

represent one-$\sigma$ standard error of the mean estimated using binning technique (reblocking) to avoid autocorrelation. For some systems, it was required to run an additional third calculation with a time-step of 0.015 a.u. to get statistical agreement within 1-$\sigma$ in the observed binding energies. In Fig. 7 we displayed the time-step convergence against the CCSD(T) reference values. Both VMC wave function optimization and FN-DMC calculations were performed with the QMeCha code[113,114,119] (version Dec22, 2024, commit b296fc0).

## Reporting summary

Further information on research design is available in the Nature Portfolio Reporting Summary linked to this article.

## Data availability

The QUID dataset generated in this study is openly available in the GitHub repository Repo-QUID with corresponding Zenodo https://doi.org/10.5281/zenodo.15880531. Source data are provided with this paper.

## Code availability

The MRCC[105,106] code and source are open-access for academic use. The QMC QMeCha package is available in a private Github repository[119], and can be requested for academic use under a CC BY-NC-ND license by contacting Dr. Matteo Barborini at matteo.barborini@uni.lu.

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

## Acknowledgements

M.P. acknowledges with gratitude financial support from the Institute for Advanced Studies (IAS) Luxembourg for the PhD "Young Academics" program. A.T. was funded by the Luxembourg Research Fund (FNR Core Grant MBD-in-BMD/18093472). The authors thank Matteo Barborini and Matteo Gori for their support and discussions, Sergio Suárez Dou for his support in biomolecular force fields, and Gregory Cordeiro Fonseca for the support in the use of his FFAST software. The financial support from the ERC Starting Grant No. 101076972, "aCCuracy", the ERC Advanced Grant No. 101054629 "FITMOL", the National Research, Development, and Innovation Office (NKFIH, Grant No. FK142489), the János Bolyai Research Scholarship of the Hungarian Academy of Sciences, and the computing time granted by the Hungarian Governmental Information-Technology Development Agency on the Komondor and the LEONARDO supercomputers are gratefully acknowledged. Some of the calculations presented in this paper were carried out using the HPC facilities of the University of Luxembourg[120] (see hpc.uni.lu). The QMC simulations were performed on the Luxembourg national supercomputer MeluXina. The authors gratefully acknowledge the LuxProvide teams for their expert support. An award of computer time was provided by the Innovative and Novel Computational Impact on Theory and Experiment (INCITE) program. This research used resources of the Argonne Leadership Computing Facility, which is a DOE Office of Science User Facility supported under Contract DE-AC02-06CH11357.

## Author contributions

M.P. generated the QUID dimers, performed DFT optimization and calculations and semiempirical, empirical, and SAPT calculations, organized and analyzed the results, and drafted the original manuscript. L.M.S. contributed to the conceptualization, manuscript writing, conducted DFT optimizations and property calculations, participated in the analysis, and served in a supervisory role. B.L. carried out the LNO-CCSD(T) calculations. J.C. performed the FN-DMC calculations and supported the SAPT calculations. D.M.R. contributed to the study design, and facilitated the procurement of computational resources through the INCITE project. P.R.N. conducted the LNO-CCSD(T) calculations, procured HPC resources, and served in a supervisory role. A.T. contributed to supervision, conceptualization, analysis, manuscript writing, and the procurement of computational resources for HPC MeluXina. All authors discussed the results and contributed to the review and editing of the article.

## Competing interests

The authors declare no competing interests.

## Additional information

¹Department of Physics and Materials Science, University of Luxembourg, Luxembourg City, Luxembourg. ²Institute for Advanced Studies, University of Luxembourg, Campus Belval, Esch-sur-Alzette, Luxembourg. ³Institute for Materials Science and Max Bergmann Center of Biomaterials, TUD Dresden University of Technology, Dresden, Germany. ⁴Department of Physical, Chemistry and Materials Science, Faculty of Chemical Technology and Biotechnology, Budapest University of Technology and Economics, Budapest, Hungary. ⁵HUN-REN-BME Quantum Chemistry Research Group, Budapest, Hungary. ⁶MTA-BME Lendület Quantum Chemistry Research Group, Budapest, Hungary. ⁷Luxembourg Researchers Hub asbl, Esch-sur-Alzette, Luxembourg. ⁸National Center for Computational Sciences, Oak Ridge National Laboratory, Oak Ridge, TN, USA. ✉e-mail: leonardo.medrano@tu-dresden.de; nagy.peter@vbk.bme.hu; alexandre.tkatchenko@uni.lu

