## [Transparent Peer Review file · Nature Communications]

Extending quantum-mechanical benchmark accuracy to biological ligand-pocket interactions

Corresponding Author: Professor Alexandre Tkatchenko

Version 0:

Reviewer comments:

Reviewer #1

(Remarks to the Author)

This work presents a QUID benchmark for 170 dimers related to ligand-protein motifs. This benchmark is valuable for developing relatively low-cost methods to study protein-ligand binding. I recommend it for publication after addressing the following comments:

1. The results suggest that sSAPT0/jaDZ performs adequately for medium-sized systems (MAE = 0.85 kcal/mol) despite its error cancellation not extending to larger systems, such as large host-guest complexes. It would be beneficial to mention the maximum error and discuss potential limitations as system size increases.
2. On Page 4, the authors state: "the overlap of the wavefunctions of the monomers within the S2 approximation." Is the S2 approximation applied only to E2_exch-ind and E2_exch-disp, or does it also affect E1_exch? By default, E1_exch in PSI4 does not use the S2 approximation. Additionally, in sSAPT0, E2_exch-ind and E2_exch-disp are scaled to minimize the impact of the S2 approximation, making it no longer a primary source of error. It would be helpful to clarify which components use the S2 approximation and how it influences the results.
3. A recent study (arXiv:2412.16405) also reports significant deviations between CCSD(T) and QMC in the S66 dataset, with a deviation of approximately 0.9 kcal/mol for the acetic acid dimer. Additionally, this deviation is linearly correlated with $\log(\text{Elst}/\text{Disp})$. It would be interesting to further explore the deviation between CCSD(T) and DMC for dimers with large and small $\log(\text{Elst}/\text{Disp})$ values in QUID, such as SF112.

Reviewer #3

(Remarks to the Author)

I have only a few remarks about this overall very interesting and well-executed study:

- * "Platinum standard" is a bit of an oversell
- * line 184: is something missing in the sentence after CHARMM36?
- * line 262: "Hirschfield" [sic] published as "F. L. Hirshfeld"
- * line 338: CPL 863, 141874 (2025) <https://doi.org/10.1016/j.cplett.2025.141874> shows clearly that CCSD(T) systematically overbinds aromatic pi stacks, albeit less strongly than suggested by FN-DMC. The latter tends to be pretty close to fully iterative CCSDT (and to the CCSD(cT) approximation considered in <http://arxiv.org/abs/2407.01442>), but the decrease in interaction energy from higher-order connected triples is partly compensated by the effect of connected quadruples at the CCSDT(Q) level. It is then to be expected that large systems like (circum)coronene dimer and "buckycatcher" in Ref. 53, which feature stacking interactions between many aromatic rings, would exhibit particularly large discrepancies between FN-DMC and localized CCSD(T).
- * line 410: a BSSE of 4% at the complete basis set limit? Not unless the word "estimated" or "extrapolated" is inserted in front of "complete". At the true CBS, BSSE should be zero, with "raw" and CP-corrected basis set limits identical. In fact, a

nontrivial residual BSSE is an indication that either the extrapolation procedure is flawed, or the basis sets used are still a bit anemic, or both. From a different perspective, one could of course exploit "raw"-CP differences as an error bar for the CBS limit.

* References: please update all references to preprints and submitted papers in the final version

Reviewer #4

(Remarks to the Author)

The topic described in the article by Puleva et al. is extremely interesting, but the manuscript presents some issues..

The main limitation lies in the justification of the choice of exchange and correlation functionals. The authors should provide more details about the choice of exchange and correlation functionals. Furthermore, since the authors present their study as a benchmark, the selection of exchange and correlation functionals is very limited. The authors should expand the number of exchange and correlation functionals and repeat the DFT calculations. For example, the authors might consider the selection and subdivision of exchange and correlation functionals proposed by Alonso-Ramos et al., *Journal of Chemical Theory and Computation* 2025, 21, 1752-1761, DOI: 10.1021/acs.jctc.4c01729.

Finally, the authors justify the choice of benzene and imidazole with the following statement: "Two small monomers were selected to represent the ligand interactions: benzene, the quintessential aromatic compound present in the phenylalanine side-chain, and imidazole, present in histidine, a more reactive and also a commonly used drug motif." However, when analyzing the structures reported in Figure S3, some structures, such as F1S3, appear difficult to generalize to other systems. The authors should provide more discussion of these results in the text. Moreover, if it is not a problem of perspective, the description of the F1I3 structure in the manuscript does not seem to be completely correct.

Version 1:

Reviewer comments:

Reviewer #1

(Remarks to the Author)

I recommend this manuscript for publication, as the authors have satisfactorily addressed all of my previous comments.

Reviewer #3

(Remarks to the Author)

I am satisfied with the revisions. The paper can now be published as far as I am concerned.

Reviewer #4

(Remarks to the Author)

The authors have responded to all the criticisms, improving the manuscript.

The selection of exchange-correlation functionals has also been expanded, even though the list is not particularly extensive.

Figure 4 of the Supporting Information concerning the F1I3 dimer, as presented in the revised version, offers a clearer depiction of the interactions involving imidazole.

Response to Reviewers: "Extending quantum-mechanical benchmark accuracy to biological ligand-pocket interactions" manuscript,
ID: NCOMMS-25-04001 in *Nature Communications*

Reviewer #1

This work presents a QUID benchmark for 170 dimers related to ligand-protein motifs. This benchmark is valuable for developing relatively low-cost methods to study protein-ligand binding. I recommend it for publication after addressing the following comments:

Our reply: We appreciate the positive review, please find point-by-point responses to the raised points below.

The results suggest that sSAPT0/jaDZ performs adequately for medium-sized systems (MAE = 0.85 kcal/mol) despite its error cancellation not extending to larger systems, such as large host-guest complexes. It would be beneficial to mention the maximum error and discuss potential limitations as system size increases.

Our reply: We thank the reviewer for their suggestion, the analysis has been carried out. The largest difference in E_{int} between LNO-CCSD(T)/CBS and sSAPT0/jaDZ is observed for the F1I1 dimer with $\Delta E_{\text{int}} = 1.97$ kcal/mol (see results for all equilibrium dimers in Fig. S3 of the SI. We also found that ΔE_{int} lies within the uncertainty estimate of the LNO-CCSD(T)/CBS method for 7 dimers, and within 1 kcal/mol for 20 dimers (for more details, see the text associated with Fig. S3 in the SI). The cases with largest discrepancies, between 1.5 kcal/mol and 2 kcal/mol, are 6 dimers of diverse sizes and geometries: F2I1 and F2I2 (54 atoms), SF1I2 (57 atoms), L4I1 (59 atoms), F1I1 (60 atoms), and SF2I3 (61 atoms), thus not indicating a relationship between ΔE_{int} and molecular size for the QUID dimers. Notably, these systems contain only imidazole as the small monomer, where the non-covalent bond has both π - π stacking character and an H-bond contribution, indicating that the increased complexity of non-covalent bonds in larger dimers presents a challenge for the sSAPT0/jaDZ method. A brief overview of the results has been added to the "Analysis of non-covalent interaction components" section, see text highlighted in blue.

On Page 4, the authors state: "the overlap of the wavefunctions of the monomers within the S2 approximation." Is the S2 approximation applied only to $E^2_{\text{exch-ind}}$ and $E^2_{\text{exch-disp}}$, or does it also affect E^1_{exch} ? By default, E^1_{exch} in PSI4 does not use the S2 approximation. Additionally, in sSAPT0, $E^2_{\text{exch-ind}}$ and $E^2_{\text{exch-disp}}$ are scaled to minimize the impact of the S2 approximation, making it no longer a primary source of error. It would be helpful to clarify which components use the S2 approximation and how it influences the results.

Our reply: We thank the reviewer for pointing out this inconsistency. Indeed, the presented sSAPT0 results were calculated with Psi4 without the S2 approximation for the E^1_{exch} term, with the S2 approximation appearing as a rescaling factor for the $E^{(20)}_{\text{exch-disp}}$ and $E^{(20)}_{\text{exch-ind}}$ in the sSAPT0 based on an empirical proportion between $E^{(10)}_{\text{exch}}$ and $E^{(10)}_{\text{exch}}(S^2)$. We have modified the corresponding text to remove the mention of the S2 approximation. Following the reviewer's suggestion, we added a sentence describing the sSAPT0 method to the "Analysis of non-covalent interaction components" section, see text highlighted in blue.

A recent study (arXiv:2412.16405) also reports significant deviations between CCSD(T) and QMC in the S66 dataset, with a deviation of approximately 0.9 kcal/mol for the acetic acid dimer. Additionally, this deviation is linearly correlated with $\log(\text{Elst}/\text{Disp})$. It would be interesting to further explore the deviation between CCSD(T) and DMC for dimers with large and small $\log(\text{Elst}/\text{Disp})$ values in QUID, such as SF1I2.

Our reply: We thank the reviewer for their suggestion. In the revised version of our manuscript, we have

Figure R 1: QUID results for the difference in the interaction energy predictions at gold benchmark levels FNO-DMC and LNO-CCSD(T) (both provided with uncertainty estimate) w.r.t the log of the Electrostatic (Elst) over the Dispersion (Disp) contributions of the sSAPT0/jaDZ analysis. The equilibrium QUID dimers are annotated in three subsets: in yellow (no H-bond in non-covalent interaction) and black (H-bond in non-covalent interaction) the ones where the predictions of the LNO-CCSD(T) and FN-DMC, $E_{predicted}$, agree within their uncertainty estimates, δE_{int} . In pink are dimers for which the same predictions do not agree within uncertainty estimates, for all of which there is an H-bond between the monomers

included the FN-DMC results for the missing dimers that were not considered in the previous version (see Fig. 3a of the main text). In this new set, the FN-DMC and LNO-CCSD(T)/CBS results for 31 out of the 42 dimers (74%) are in agreement within uncertainty estimates. One should keep in mind that on Fig. 3b of the manuscript, ΔE_{int} is given as the difference between the predictions. However, the highest discrepancy, taken as the minimum distance between the predictions taking into account their uncertainty estimates (i.e. as a difference between the ends of the error bars) is 0.83 kcal/mol for the H-bonded L2I2 dimer (see Fig. S4 of the SI).

The distribution of the discrepancies in the E_{int} predictions between FN-DMC and LNO-CCSD(T)/CBS w.r.t. $\log(\text{Elst}/\text{Disp})$ are consistent with the findings for the S66x8 dataset from arXiv:2412.16405, in particular for the cases of acetic dimer and our H-bonded SF1I2 - see Fig. R1 and Fig. 3b of the manuscript. When the electrostatic contribution is the distinctly dominant term (always with an H-bond present in the dimer), this also results in a higher discrepancy between the two gold standard methods. We have added a discussion of these results in the "Analysis of non-covalent interaction components" section, see text highlighted in blue.

The results in the paper have been updated based on the new FN-DMC data, but the main conclusions remain unchanged. The updated MAE of FN-DMC w.r.t LNO-CCSD(T)/CBS is 0.47 kcal/mol, RMSE 0.60 kcal/mol, while the mean uncertainties of the LNO-CCSD(T) and FN-DMC methods are both 0.38 kcal/mol. The "Towards "platinum standard" in E_{int} by benchmarking "gold standard" methods LNO-CCSD(T) and FN-DMC" section in the manuscript has been modified accordingly to include the new results and analysis, see Fig. 3a of the manuscript.

Reviewer #3

I have only a few remarks about this overall very interesting and well-executed study:

Our reply: We appreciate the positive review, please find point-by-point responses to the raised points below.

"Platinum standard" is a bit of an oversell

Our reply: We thank the reviewer for their comment. While our use of the term "platinum" has been defined in the paper to highlight the extra level of verification within the dataset creation involving comparison of two golden benchmarks for computing interaction energies in diverse ligand-pocket motifs, and is used in quotation marks accordingly, we understand the concern raised by the reviewer. The "platinum" standard has been accordingly removed from the abstract, where it was not defined, and has been replaced by "QUID goes beyond the "gold standard"", please find the change in the abstract highlighted in blue text.

line 184: is something missing in the sentence after CHARMM36?

Our reply: We would like to thank the reviewer for their comment, the sentence has now been modified for clarity. The new sentence denoted in blue reads: "The most prominent outliers are found for the methods CHARMM36 and DFTB3+MBD, with E_{int} values in ranges from -12.5 kcal/mol to -5 kcal/mol (details in Fig. S6 of the SI) and -7.5 kcal/mol to -4.5 kcal/mol, correspondingly."

line 262: "Hirschfield" [sic] published as "F. L. Hirshfeld"

Our reply: We would like to thank the reviewer for their comment, the typos have been fixed, changes highlighted in blue.

line 338: CPL 863, 141874 (2025) <https://doi.org/10.1016/j.cplett.2025.141874> shows clearly that CCSD(T) systematically overbinds aromatic pi stacks, albeit less strongly than suggested by FN-DMC. The latter tends to be pretty close to fully iterative CCSDT (and to the CCSD(cT) approximation considered in <http://arxiv.org/abs/2407.01442>), but the decrease in interaction energy from higher-order connected triples is partly compensated by the effect of connected quadruples at the CCSDT(Q) level. It is then to be expected that large systems like (circum)coronene dimer and "buckycatcher" in Ref. 53, which feature stacking interactions between many aromatic rings, would exhibit particularly large discrepancies between FN-DMC and localized CCSD(T).

Our reply: We thank the reviewer for raising this important point from these two most recent studies. Combining our response to the last question of reviewer 1 and this comment from reviewer 2, we extended the discussion on the similarities and differences between the CC-DMC comparison in the two noted works and the present case of the QUID systems as follows:

"From this perspective, the QUID systems differ from the supramolecular complexes with extended π - π interactions, where some of us uncovered considerable disagreements between CCSD(T) and FN-DMC.⁵⁵ As noted in Ref. 55 and more recent studies,^{56,63-65} e.g., the FN approximation, time-step discretization, pseudopotential, post-CCSD(T) and other high-order effects could be notable for extended π - π interactions. However, beyond CCSD(T) corrections are shown to be negligible for our purposes in H-bonded dimers.^{63,65}

line 410: a BSSE of 4% at the complete basis set limit? Not unless the word "estimated" or "extrapolated" is inserted in front of "complete". At the true CBS, BSSE should be zero, with "raw" and CP-corrected basis set limits identical. In fact, a nontrivial residual BSSE is an indication that either the extrapolation procedure is flawed, or the basis sets used are still a bit anemic, or both. From a different perspective, one could of course exploit "raw"-CP differences as an error bar for the CBS limit.

Our reply: We thank the reviewer for their comment. We agree with this point on phrasing and the word 'extrapolated' has been added in the manuscript accordingly, change denoted in blue. As we document in detail in Fig. S20 of the SI, CP corrected aQZ, CBS(aD,aT) and CBS(aT,aQ) results are very close, showing excellent convergence with CP, even if the "raw", CP-uncorrected results are not as converged. We also considered the suggested "raw"-CP difference as the basis set uncertainty component. However, for the above reason this would considerably overestimate the basis set incompleteness compared to the uncertainty measure based on aXZ-CBS[a(X-1),aX], that we use in the Methods section.

References: please update all references to preprints and submitted papers in the final version

Our reply: We thank the reviewer for their comment, the references now correspond to published papers and have been updated accordingly.

Reviewer #4

The topic described in the article by Puleva et al. is extremely interesting, but the manuscript presents some issues.

Our reply: We appreciate the feedback, please find point-by-point responses to the raised points below.

The main limitation lies in the justification of the choice of exchange and correlation functionals. The authors should provide more details about the choice of exchange and correlation functionals. Furthermore, since the authors present their study as a benchmark, the selection of exchange and correlation functionals is very limited. The authors should expand the number of exchange and correlation functionals and repeat the DFT calculations. For example, the authors might consider the selection and subdivision of exchange and correlation functionals proposed by Alonso-Ramos et al., Journal of Chemical Theory and Computation 2025, 21, 1752-1761, DOI: 10.1021/acs.jctc.4c01729.

Our reply: We thank the reviewer for their suggestion. The interaction energy of the QUID dimers has now been calculated with additional DFT functionals selected by cross-referencing the list from Ref. 1 Alonso-Ramos et al. (Journal of Chemical Theory and Computation 2025, 21, 1752-1761, DOI:10.1021/acs.jctc.4c01729) and the study on non-covalent systems Ref. 2 "Density functional theory for van der Waals complexes: Size matters" by M. Gray and J. M. Herbert (Annual Reports in Computational Chemistry 2024, DOI:10.1016/bs.arcc.2024.03.001). Namely, the additional DFT functionals we have considered are: the GGA functional PBE-MBD; the global hybrids BH&HLYP-XDM and B3LYP-D3; the range-separated hybrids ω B97X-V and CAM-B3LYP-XDM; and the double hybrid functional PBE-QIDH-D3.

The results indicated consistently good performance for the ω B97X functional with both VV10 and D3 dispersion corrections. The newly investigated ω B97X-V functional proved to be one of the best in equilibrium and the short-ranged *compressed* regime for the non-equilibrium dimers, while also performing well in the *elongated* regime. Notably, the double hybrid functional PBE-QIDH-D3 results in a higher MAE value for the equilibrium QUID dimers than its counterparts PBE0-MBD and PBE-MBD, likely due to the MP2 corrections drawback in dispersion-dominated large non-covalent complexes (see Fig. R2 and Fig. 4 of the main text). This result, the cost and efficacy of double hybrids as stated in Ref. 2, and the role of the parametrization of dispersion corrections for such functionals geared towards smaller conformers and dimers systems (Physical Chemistry Chemical Physics, 2017, 19(21), 13481-13487, DOI: <https://doi.org/10.1039/C7CP00709D>) suggest that range-separated double-hybrid functionals are not well-suited for such large non-covalent dimers and were therefore not investigated further. The text in "Assessing the performance of DFT, semiempirical, and empirical methods" section, as well as Figs. 4 and 5 of the manuscript, have been modified to include the new results. Related figures and tables in the SI have also been updated. The text discussing the new results has been highlighted in blue.

Finally, the authors justify the choice of benzene and imidazole with the following statement: "Two small monomers were selected to represent the ligand interactions: benzene, the quintessential aromatic compound present in the phenylalanine side-chain, and imidazole, present in histidine, a more reactive and also a commonly used drug motif." However, when analyzing the structures reported in Figure S3, some structures, such as FIS3, appear difficult to generalize to other systems. The authors should provide more discussion of these results in the text. Moreover, if it is not a problem of perspective, the description of the F1I3 structure in the manuscript does not seem to be completely correct.

Our reply: We thank the reviewer for their comment. The QUID dimers are comprised of monomers interacting in one or more of the three most frequent interaction types appearing between ligands and side chain groups of the amino acids in a pocket, that is aliphatic-aromatic, H-bonding, and π -stacking. For example, the folded F1I3 mimics a more crowded binding pocket (see example in Ref: M. Gao, PLOS Computational Biology

Figure R 2: **Comparison of interaction energy predictions using DFT, semiempirical, and empirical methods with respect to LNO-CCSD(T) and to each other.** **a** Distributions of interaction energy predictions w.r.t. LNO-CCSD(T), ΔE_{int} , showed via box plots, for a selection of computational methods - DFT methods: PBE0+MBD, ω B97X-V, PBE0+D4, PBE-MBD, ω B97X-D3, PBE0+XDM, PBE-QIDH-D3, PBE0+MBD-NL, B3LYP-D3, PBE0+TS; semiempirical methods: DFTB3+MBD, GFN2-xTB; and classical FFs: AMBER-GAFF2 and CHARMM36. The negative ΔE_{int} values signify underbinding, while the positive ones overbinding. **b** A heatmap of MAE values of predicted E_{int} w.r.t LNO-CCSD(T) for the QUID equilibrium dimers in the first column, and the MAE of all methods w.r.t each other in subsequent columns. The computational methods to predict E_{int} were the same methods as in panel a. *For the LNO-CCSD(T) method, the value shown with asterisk is the mean absolute of the uncertainty estimates for E_{int} .

9(10), e1003302, (2013)), stabilised by aliphatic-aromatic interactions and an H-bond with an imidazole in a histidine group. Analogously, a linear L2B1 dimer represents a toy model of a more open surface pocket (see example in Ref: D. K. Johnson and Karanicolas, PLoS computational biology 9, e1002951, (2013)). Following this discussion, additional characterization details have been added to the description of the dimer structure generation in the "Quantum-mechanical exploration of binding interactions" section of the Results. The representation of the F1I3 dimer in Fig. 4 of the SI has been updated for clarity as well.